# Scale Equalization for Multi-Level Feature Fusion

**Bum Jun Kim**                                                    *kmbmjn@postech.edu*
*Department of Electrical Engineering*
*Pohang University of Science and Technology*

**Sang Woo Kim**                                                   *swkim@postech.edu*
*Department of Electrical Engineering*
*Pohang University of Science and Technology*

**Reviewed on OpenReview:** *https://openreview.net/forum?id=oS4SkVKA7S*

## Abstract

Deep neural networks have exhibited remarkable performance in a variety of computer vision fields, especially in semantic segmentation tasks. Their success is often attributed to multi-level feature fusion, which enables them to understand both global and local information from an image. However, multi-level features from parallel branches exhibits different scales, which is a universal and unwanted flaw that leads to detrimental gradient descent, thereby degrading performance in semantic segmentation. We discover that scale disequilibrium is caused by bilinear upsampling, which is supported by both theoretical and empirical evidence. Based on this observation, we propose injecting scale equalizers to achieve scale equilibrium across multi-level features after bilinear upsampling. Our proposed scale equalizers are easy to implement, applicable to any architecture, hyperparameter-free, implementable without requiring extra computational cost, and guarantee scale equilibrium for any dataset. Experiments showed that adopting scale equalizers consistently improved the mIoU index across various target datasets, including ADE20K, PASCAL VOC 2012, and Cityscapes, as well as various decoder choices, including UPerHead, PSPHead, ASPPHead, SepASPPHead, and FCNHead.

## 1 Introduction

Deep neural networks have shown remarkable performance, especially in the computer vision field. Their substantial modeling capability has enabled us to develop significantly accurate models with rich image features for a wide range of vision tasks, including object detection and semantic segmentation.

One challenge in computer vision tasks is understanding both the global and local contexts of an image (Reddi et al., 2018; Tu et al., 2022). Indeed, the cascade architecture of a deep neural network faces difficulty in understanding multiple contexts of an image owing to the single-level feature it uses. To address this problem, modern vision networks have employed a parallel architecture that aggregates multi-level features in different spatial sizes to extract both global and local information from an image. For semantic segmentation as an example, multi-level feature fusion has been adopted in numerous models such as UPerNet (Xiao et al., 2018), PSPNet (Zhao et al., 2017), DeepLabV3 (Chen et al., 2017), DeepLabV3+ (Chen et al., 2018b), FCN (Long et al., 2015), and U-Net (Ronneberger et al., 2015).

Although the parallel architecture builds multiple fastlanes to facilitate multi-level features to contribute to output, if certain features are not involved in the fusion, they simply waste computational resources. Initially, all feature branches should be exploited to explore their potential usefulness, and after training, their optimal combination should be obtained. Thus, the underlying assumption of multi-level feature fusion is that, at least in an initialized state, all multi-level features will participate in producing a fused feature. However, we claim that existing architectural design for multi-level feature fusion in semantic segmentation has a potential problem of scale disequilibrium, which yields unwanted bias that diminishes the contribution

of several features. Specifically, the multi-level features exhibit different scales at initialization, which leads to different contributions and gradient scales, thereby hindering training with gradient descent. We identify the cause of the scale disequilibrium—bilinear upsampling, which is used to enlarge multi-level features to the same spatial size. Demonstration of the scale disequilibrium caused by bilinear upsampling is provided theoretically and empirically.

To solve the scale disequilibrium problem, this study proposes injecting scale equalizers into multi-level feature fusion. The scale equalizer normalizes each feature using the global mean and standard deviation, which guarantees scale equilibrium across multi-level features. Because the proposed scale equalizer is simply global normalization, which uses empirical values for subtraction and division, its implementation is easy and hyperparameter-free, requires little extra computation that is actually free, and assures scale equilibrium for any datasets and architectures. Experiments on semantic segmentation tasks showed that applying scale equalizers for multi-level feature fusion consistently improved the mIoU index across extensive experimental setups, including datasets and backbones.

## 2 Background

**Formulation**   This study considers the standard framework for supervised learning of semantic segmentation networks because it is a representative task using multi-level feature fusion (Fig. 1). Let $\mathbf{I} \in \mathbb{R}^{H \times W \times C}$ be an input image to a semantic segmentation network, where $H, W$ is the size of the image and $C$ represents the number of image channels. The objective of semantic segmentation is to generate a semantic mask $\hat{\mathbf{Y}} \in \mathbb{R}^{H \times W \times N_c}$ that classifies each pixel in the image $\mathbf{I}$ into one of the $N_c$ categories. A deep neural network, which comprises an encoder and decoder, is employed as a semantic segmentation model that outputs $\hat{\mathbf{Y}}$ from the input image $\mathbf{I}$. The encoder is a backbone network with several stages where the input image first goes through. The decoder—also referred to as the head—uses a set of intermediate feature maps $\{\mathbf{C}_i\}$ from the encoder to produce the segmentation output $\hat{\mathbf{Y}}$. To quantify a difference between the prediction $\hat{\mathbf{Y}}$ and the ground truth $\mathbf{Y}$, a loss function such as pixel-wise cross-entropy is used. With gradient descent optimization for the loss function, the encoder and decoder are trained together on an image-label pair dataset by the fine-tuning strategy, where the encoder begins with a pretrained weight whereas the decoder is trained from scratch.

### 2.1 Multi-Stage Feature Fusion

The last feature map of the encoder contains rich, high-level information on the image (Chen et al., 2018a) and is included in the set of feature maps used by the decoder. However, each stage of the encoder yields a downsampled feature map. Thus, the last feature map exhibits a severe downsampling rate, losing fine details in the image (Chen et al., 2017). To address this problem, modern decoders have used multiple feature maps from several stages to aggregate information with various spatial sizes (Kirillov et al., 2019; Zheng et al., 2021). We refer to this practice as *multi-stage feature fusion*. For multi-stage feature fusion, the common choice on the set of features is to use the encoder outputs of the second to fifth stages, *i.e.*, $\{\mathbf{C}_2, \mathbf{C}_3, \mathbf{C}_4, \mathbf{C}_5\}$ (Lin et al., 2017). The use of the first stage output is commonly avoided because it requires large GPU memory. For convolution-based residual networks (He et al., 2016) and certain vision transformers such as Swin (Liu et al., 2021), the downsampling ratios of the four encoder features are $\{4, 8, 16, 32\}$. Others, such as vanilla vision transformers (Dosovitskiy et al., 2021), keep the same spatial size for the four encoder features. Despite the promising results of the latter, in general vision tasks, progressive downsampling is critical to encouraging heterogeneous characteristics in multiple feature maps, which advocates the former networks. This study targets the former and describes a problem regarding feature fusion using different downsampling ratios. The remainder of this section reviews the detailed mechanism of a modern decoder with multi-stage feature fusion.

**UPerHead**   UPerHead, the head deployed in UPerNet (Xiao et al., 2018), is a prime example of a decoder designed for multi-stage feature fusion. Recent vision transformers have preferred the UPerHead for semantic segmentation tasks (Dosovitskiy et al., 2021; Bao et al., 2022; He et al., 2022), and their remarkable performance let it be one of the most widely used decoders in the current state. The UPerHead comprises different

Table 1: List of notations used in this study.

| Notation | Meaning |
|---|---|
| $\mathbf{I}$ | An input image to a semantic segmentation network. |
| $H, W, C$ | Height, width, and the number of channels for the input image. |
| $\hat{\mathbf{Y}}$ | Predicted result for semantic segmentation. |
| $N_c$ | The number of classes to be classified in the semantic segmentation task. |
| $\{\mathbf{C}_i\}$ | A set of intermediate feature maps $\{\mathbf{C}_i\}$ from the encoder. |
| $\{\mathbf{L}_i\}$ | Laterals UPerHead. |
| $\{\mathbf{F}_i\}$ | Outputs of the top-down pathway of the FPN. |
| $\{\mathbf{P}_i\}$ | Outputs of the FPN, which is concatenation subjects with optional bilinear upsampling. |
| $\mathrm{UP}_r$ | $r\times$ blinear upsampling. |
| $r$ | Upsampling ratio in bilinear upsampling. |
| $h$ | A convolutional unit block. |
| $\mathbf{Z}$ | Fused feature after multi-level feature fusion. |
| $s$ | Output stride, *i.e.*, downsampling ratio at the last stage. |
| $n_b$ | The number of branches in multi-level feature fusion. |
| $x_i$ | $i$th concatenation subject in multi-level feature fusion. |
| $w_i$ | $i$th weight of the linear layer in multi-level feature fusion. |
| $b$ | Bias in the linear layer in multi-level feature fusion. |
| $y$ | The fused feature $y = \sum_i w_i x_i + b$ in multi-level feature fusion. |
| $\frac{\partial y}{\partial w_i}$ | Partial derivative of $y$ with respect to $w_i$. |
| $\mathbb{E}[\mathbf{x}]$ | Mean of a feature $\mathbf{x}$. |
| $\mathrm{Var}[\mathbf{x}]$ | Variance of a feature $\mathbf{x}$. |
| $L$ | Loss function such as the pixel-wise cross-entropy loss function. |
| $\eta$ | Learning rate used in gradient descent. |
| $\gamma$ | Scale term in batch normalization, which is initialized to one. |
| $\beta$ | Shift term in batch normalization, which is initialized to zero. |
| $\mathbf{W}$ | Weight matrix. |
| $\mathbf{x}$ | Feature vector. |
| ReLU | Rectified Linear Unit as $\mathrm{ReLU}(x) = \max(0, x)$. |
| BatchNorm | Batch normalization operation. |
| $\mathbf{W}^{\mathrm{fuse}}$ | Weight matrix of the convolutional layer in the convolutional unit block of fusion. |
| $\mathbf{Z}^{\mathrm{fuse}}$ | An intermediate result that is obtained after the convolutional layer of fusion with $\mathbf{W}^{\mathrm{fuse}}$. |
| $\mathbf{X}$ | Arbitrary feature. |
| $\mathcal{N}(\mu, \sigma^2)$ | Normal distribution with mean $\mu$ and variance $\sigma^2$. |

modules, such as the pyramid pooling module (PPM) (Zhao et al., 2017), feature pyramid network (FPN) (Lin et al., 2017), and convolutional unit block, which is composed of convolution, batch normalization, and ReLU operation. Firstly, each of the three feature maps $\{\mathbf{C}_2, \mathbf{C}_3, \mathbf{C}_4\}$ is subjected to a convolutional unit block to yield laterals $\{\mathbf{L}_2, \mathbf{L}_3, \mathbf{L}_4\}$. Additionally, the last lateral $\mathbf{L}_5$ is produced from the last feature map $\mathbf{C}_5$ using the PPM module that is described in Section 2.2. Now, a subnetwork called FPN performs the top-down pathway to laterals to obtain its output $\mathbf{F}_i = \mathbf{L}_i + \mathrm{UP}_2(\mathbf{F}_{i+1})$ for $i \in \{2, 3, 4\}$ and $\mathbf{F}_5 = \mathbf{L}_5$, where the operation $\mathrm{UP}_r$ denotes $r\times$ bilinear upsampling (Appendix A). Subsequently, FPN applies a convolutional unit block $h_i$ to each result to yield its output $\mathbf{P}_i = h_i(\mathbf{F}_i)$ for $i \in \{2, 3, 4, 5\}$. The set of FPN output $\{\mathbf{P}_2, \mathbf{P}_3, \mathbf{P}_4, \mathbf{P}_5\}$ has the same spatial size as encoder features $\{\mathbf{C}_2, \mathbf{C}_3, \mathbf{C}_4, \mathbf{C}_5\}$, keeping their downsampling ratios $\{4, 8, 16, 32\}$. Thus, feature fusion for FPN outputs requires $2^{i-2}\times$ bilinear upsampling for each $\mathbf{P}_i$ to ensure that they share the same spatial size of $H/4 \times W/4$. Finally, they can be concatenated together with respect to channel dimension and fused with a convolutional unit block $h$ as

$$\mathbf{Z} = h([\mathbf{P}_2; \mathrm{UP}_2(\mathbf{P}_3); \mathrm{UP}_4(\mathbf{P}_4); \mathrm{UP}_8(\mathbf{P}_5)]). \tag{1}$$

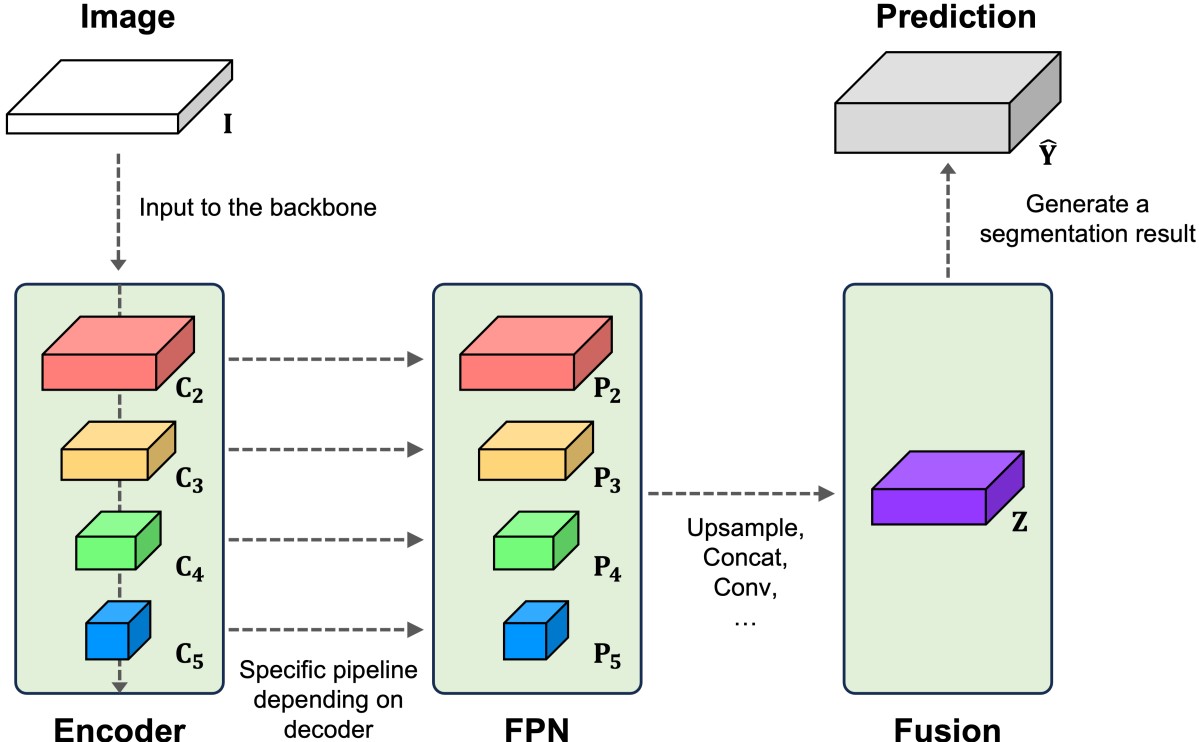

Figure 1: An overview of a semantic segmentation network. Input image $\mathbf{I}$ is fed to the backbone to yield encoder features $\{\mathbf{C}_i\}$. They are used to obtain FPN outputs $\{\mathbf{P}_i\}$, which are fused through upsampling, concatenation, convolution, etc. Finally, a segmentation result $\hat{\mathbf{Y}}$ is generated using the fused feature $\mathbf{Z}$.

The fused feature $\mathbf{Z}$ is then subjected to a $1\times1$ convolution and a $4\times$ bilinear upsampling to yield a predicted semantic mask $\hat{\mathbf{Y}}$, which has the size of $H \times W \times N_c$.

## 2.2 Single-Stage Feature Fusion

Although multi-stage feature fusion uses a set of encoder features from several stages, certain decoders such as PSPHead (Zhao et al., 2017) or ASPPHead (Chen et al., 2017) only use a single feature map from the last stage $\mathbf{C}_5$. They modify the encoder to exhibit a downsampling ratio of 8 or 16 at the last stage, which is referred to as the output stride. Denoting the output stride as $s$, the spatial size of the last feature map $\mathbf{C}_5$ is $(H/s) \times (W/s)$. These decoder heads perform dissimilar feature fusion: From a single-stage feature, multiple feature maps with various sizes are produced, which are then fused. We refer to this type of feature fusion as *single-stage feature fusion*. In a similar but different way, single-stage feature fusion enables the decoder to extract both global and local contexts from the targeted feature map. The remainder of this section reviews the detailed mechanisms of modern decoders with single-stage feature fusion.

**PSPHead** PSPHead refers to the head deployed in PSPNet (Zhao et al., 2017). Its underlying mechanism is to extract global and local contexts from a feature map using multiple branches, which is referred to as a pyramid pooling module (PPM). Targeting the last feature map $\mathbf{C}_5$, it performs four average poolings in parallel, which yields features with spatial sizes $1\times1$, $2\times2$, $3\times3$, and $6\times6$. Subsequently, a convolutional unit block is applied to each branch, and then each result is upsampled to fit the size of $\mathbf{C}_5$. Along with the feature map $\mathbf{C}_5$, the results from the four branches are concatenated together with respect to channel dimension. Finally, a convolutional unit block $h$ is applied to fuse them. Denoting the outputs of convolutional unit

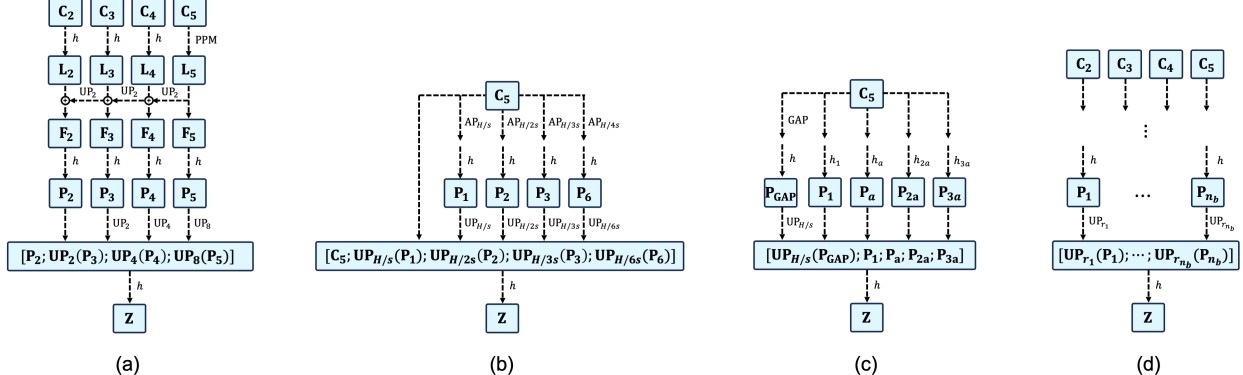

Figure 2: Visualization of the architecture of modern decoders: (a) UPerHead, (b) PSPHead, (c) ASPPHead and SepASPPHead, and (d) their general form.

blocks in parallel branches as $\{\mathbf{P}_1, \mathbf{P}_2, \mathbf{P}_3, \mathbf{P}_6\}$, fusing them is represented as

$$\mathbf{Z} = h([\mathbf{C}_5; \mathrm{UP}_{H/s}(\mathbf{P}_1); \mathrm{UP}_{H/2s}(\mathbf{P}_2); \mathrm{UP}_{H/3s}(\mathbf{P}_3); \mathrm{UP}_{H/6s}(\mathbf{P}_6)]), \tag{2}$$

where $H = W$ is assumed for notational simplicity. Similarly, the fused feature $\mathbf{Z}$ is then subjected to a $1 \times 1$ convolution and a $4\times$ bilinear upsampling to yield a predicted semantic mask $\hat{\mathbf{Y}}$, which has the size of $H \times W \times N_c$.

**ASPPHead and Others** ASPPHead refers to the head deployed in DeepLabV3 (Chen et al., 2017). It uses atrous convolution (Yu & Koltun, 2016; Chen et al., 2018a), which generates empty space between each element of the convolutional kernel. To extract both global and local information from a feature map, the ASPPHead adopts multiple atrous convolutions with various atrous rates in parallel. For the last feature map $\mathbf{C}_5$, the first branch applies a series of global average pooling (GAP), convolutional unit block, and bilinear upsampling to restore the spatial size prior to the GAP. Each of the other four branches applies a convolutional unit block whose convolutional operation adopts an atrous rate $\{1, a, 2a, 3a\}$, where $a = 96/s$. The results from the five branches are concatenated together with respect to channel dimension, and then a convolutional unit block $h$ is applied to fuse them. Denoting the outputs of convolutional unit blocks in parallel branches as $\{\mathbf{P}_{\mathrm{GAP}}, \mathbf{P}_1, \mathbf{P}_a, \mathbf{P}_{2a}, \mathbf{P}_{3a}\}$, fusing them is represented as

$$\mathbf{Z} = h([\mathrm{UP}_{H/s}(\mathbf{P}_{\mathrm{GAP}}); \mathbf{P}_1; \mathbf{P}_a; \mathbf{P}_{2a}; \mathbf{P}_{3a}]). \tag{3}$$

Similarly, the fused feature $\mathbf{Z}$ is then subjected to a $1 \times 1$ convolution and a $s\times$ bilinear upsampling to yield a predicted semantic mask $\hat{\mathbf{Y}}$, which has the size of $H \times W \times N_c$. In DeepLabV3+ (Chen et al., 2018b), a variant called SepASPPHead is developed using depthwise separable convolutions instead, while keeping the same decoder architecture. This single-stage feature fusion has also been used in other segmentation networks such as FCN (Long et al., 2015) and U-Net (Ronneberger et al., 2015), which progressively repeats fusion for two features with upsampling at each time.

**Summary and Generalization** As reviewed above, modern decoders of segmentation networks perform multi- or single-stage feature fusion, which we collectively refer to as *multi-level feature fusion*. Although each decoder has a distinct architecture, their feature fusions share a similar design pattern (Fig. 2). Using single or multiple encoder features, certain operations are applied in parallel branches, and the convolutional unit block in the $i$th branch generates the $i$th feature map $\mathbf{P}_i$ for $i \in \{1, \cdots, n_b\}$ for the number of branches $n_b$. Because the spatial size of each feature map $\mathbf{P}_i$ differs, optional $r_i\times$ bilinear upsampling is needed to assure the same spatial size. For notational simplicity, $1\times$ bilinear upsampling is defined as the identity operation. Because the set of encoder features for fusion includes a feature map that does not require bilinear upsampling, at least one branch exhibits the upsampling ratio $r_i = 1$, whereas others use $r_i > 1$. The fused

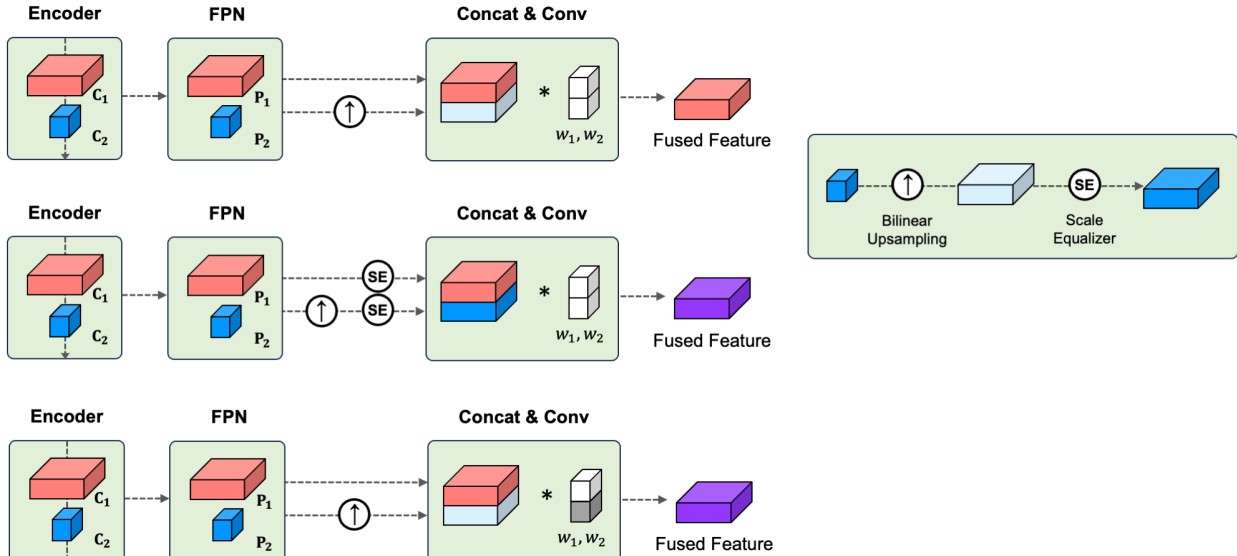

Figure 3: Overview of the problem statement and the proposed solution. This illustration depicts a fusion by UPerHead for two features for simplicity, but nonetheless, the common fusion scheme uses four features. (Top) Existing multi-level feature fusion concatenates features after bilinear upsampling. The variances of concatenation subjects, represented as chroma in this figure, exhibit disequilibrium because bilinear upsampling decreases variance. In this fusion, $\mathbf{P}_1$ dominates in the fused feature as a red color, which diminishes the contribution of $\mathbf{P}_2$ and causes slower training on $w_2$. (Middle) Our proposed multi-level feature fusion with scale equalizers guarantees consistent variance across subjects of concatenation. In this scheme, a suitably fused feature as a purple color is produced with alive gradients with respect to both $w_1$ and $w_2$. (Bottom) Efficient implementation of our proposed method, where scale equalizers are replaced by applying auxiliary initialization for $w_1$ and $w_2$.

feature is now obtainable by concatenation with respect to channel dimension and a convolutional unit block $h$ as

$$\mathbf{Z} = h([\mathrm{UP}_{r_1}(\mathbf{P}_1); \cdots ; \mathrm{UP}_{r_{n_b}}(\mathbf{P}_{n_b})]). \tag{4}$$

Below, we investigate the concatenation subjects $\mathrm{UP}_{r_i}(\mathbf{P}_i)$. Although we introduced the encoder features $\{\mathbf{C}_i\}$ for detailed descriptions of decoders, they will not be further used in our analysis.

## 3 Scale Disequilibrium

### 3.1 Problem Statement

As reviewed above, the decoder of the segmentation network includes a module to fuse features of varied sizes. Here, we claim that multi-level feature fusion requires explicit scale equalization because they exhibit different scales, which causes scale disequilibrium on gradients (Fig. 3).

To understand feature scale, this study uses feature variance. Other measures such as the norm depend on the size of the feature, whereas variance provides a suitably scaled result with respect to its size. Owing to the effectiveness of variance in understanding feature scales, it has been adopted in several pieces of literature (Glorot & Bengio, 2010; He et al., 2015; Klambauer et al., 2017). We also employ the mean of a feature to understand its representative value as occasion arises. Using variance, we describe the scale disequilibrium as follows.

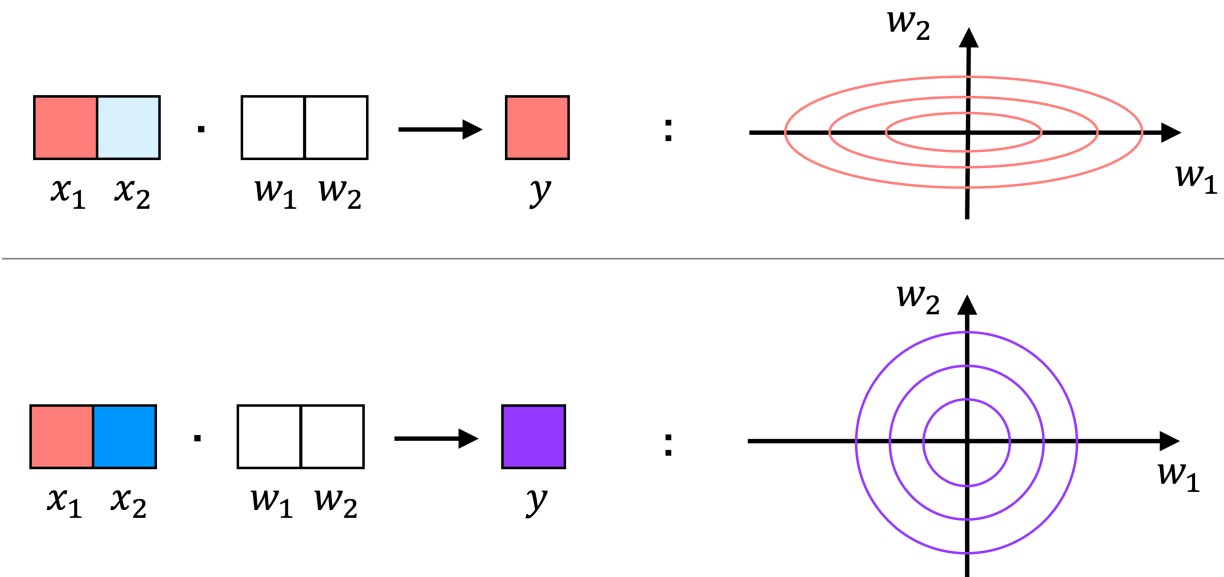

Figure 4: Illustration of scale disequilibrium. (Top) When $\mathrm{Var}[x_1] > \mathrm{Var}[x_2]$, we obtain $\mathrm{Var}[\frac{\partial y}{\partial w_1}] > \mathrm{Var}[\frac{\partial y}{\partial w_2}]$, whose landscape is difficult to optimize through gradient descent. (Bottom) Achieving scale equilibrium $\mathrm{Var}[x_1] = \mathrm{Var}[x_2]$ stabilizes the landscape and corresponding gradient descent optimization with respect to $w_1$ and $w_2$.

**Proposition 3.1.** *Consider a multi-level feature fusion, where a concatenated feature $[x_1; x_2]$ is subjected to a linear layer with weight $[w_1, w_2]$ and bias $b$ to yield the fused feature $y = w_1 x_1 + w_2 x_2 + b$. When the two features $x_1$ and $x_2$ are on different scales, i.e., $\mathrm{Var}[x_1] \neq \mathrm{Var}[x_2]$, the gradients of the fused feature with respect to the corresponding weight exhibit scale disequilibrium, i.e., $\mathrm{Var}[\frac{\partial y}{\partial w_1}] \neq \mathrm{Var}[\frac{\partial y}{\partial w_2}]$.*

The proof is straightforward because $\frac{\partial y}{\partial w_i} = x_i$. From the chain rule, the gradient of a loss function $L$ with respect to weight $w_i$ is $\frac{\partial L}{\partial w_i} = \sum_y \frac{\partial L}{\partial y} \frac{\partial y}{\partial w_i}$, and thus the gradient scale is affected by the scale of the corresponding feature $x_i$. The term linear layer indicates a fully connected layer or a convolutional layer.

For example, consider scale disequilibrium for concatenation subjects where $\mathrm{Var}[x_1] = 10\mathrm{Var}[x_2]$. Then we obtain $\mathrm{Var}[\frac{\partial y}{\partial w_1}] = 10\mathrm{Var}[\frac{\partial y}{\partial w_2}]$, and thus gradient descent on $w_2$ is on a ten times smaller scale than $w_1$, which slows down the training on $w_2$ (Fig. 4). However, gradient descent optimizers inherently assume scale equilibrium on gradients (Zeiler, 2012): For gradient descent $w_i \leftarrow w_i - \eta \frac{\partial L}{\partial w_i}$, the weight initializer sets the same scale of initial weight $\mathrm{Var}[w_i]$ for weights within the same linear layer, and common gradient descent uses a single learning rate $\eta$ without scale discrimination, which leads to difficulty in capturing different gradient scales $\mathrm{Var}[\frac{\partial L}{\partial w_i}]$. Note that existing literature (Glorot & Bengio, 2010; He et al., 2015; Klambauer et al., 2017; Bachlechner et al., 2021) have discussed matching gradient scales across inter-layers to ensure stable gradient descent without poor training dynamics such as vanishing or exploding gradients. On top of inter-layer gradient scales, we claim to equalize intra-layer gradient scales. For the feature fusion scenario, matching the intra-layer gradient scales $\mathrm{Var}[\frac{\partial y}{\partial w_1}] = \mathrm{Var}[\frac{\partial y}{\partial w_2}]$ requires scale equalization for the subjects of concatenation: $\mathrm{Var}[x_1] = \mathrm{Var}[x_2]$. Achieving scale equilibrium eliminates the hidden factor that causes degradation in gradient descent optimization, which enhances the training of the segmentation network as well as the performance of semantic segmentation.

Furthermore, the gradient scale indicates the amount of contribution: A smaller scale on the gradient or feature indicates less contribution to the predicted mask. For example, when the last feature map that contains rich, high-level image information contributes little to the predicted mask, the quality of the segmentation result would degrade. To use the last feature map while supplementing its deficient information using multi-level features, it is desirable to ensure a balanced contribution from the multi-level features. Note

that we are not saying that the amount of feature contributions should be precisely controlled to be optimal at initialization; rather, we would like to equalize feature contributions at the initial state and then let them change to be optimal during training. Our claim is that unwanted imbalances in gradient scales should be avoided at initialization. This claim is supported by the above existing literature on matching inter-layer gradient scales, which have approached it this way and emphasized avoiding unwanted imbalances such as vanishing or exploding gradients at the initial state; thereafter, gradient scales are allowed to change during training. Considering this objective, our argument can be interpreted as establishing a valid initialization to achieve scale equilibrium on gradients with respect to intra-layer weights.

These arguments can be extended to match the mean of gradients, which requires the same mean for the subjects of concatenation: $\mathbb{E}[x_1] = \mathbb{E}[x_2]$. Based on this claim, we inspect the scale of concatenation subjects in the modern decoder of the segmentation network.

**Batch Normalization Partially Equalizes Scale**  Fortunately, the use of batch normalization results in a normalized feature[1] and thus concatenation of several features from the output of batch normalization is allowable. Moreover, batch normalization allows the use of convolution with arbitrary weights $\mathbf{W}$ and ReLU operation without causing scale disequilibrium. This is because the output of a convolutional unit block with the pipeline [Conv–BatchNorm–ReLU] yields a fixed mean and variance without requiring any specific conditions on the weight $\mathbf{W}$ and feature $\mathbf{x}$:

$$\mathbb{E}[\text{ReLU}(\text{BatchNorm}(\mathbf{W}\mathbf{x}))] = \frac{1}{\sqrt{2\pi}}, \tag{5}$$

$$\text{Var}[\text{ReLU}(\text{BatchNorm}(\mathbf{W}\mathbf{x}))] = \frac{\pi - 1}{2\pi}. \tag{6}$$

This property also implies that any architecture can be freely chosen before the input of the convolutional unit block. Furthermore, batch normalization guarantees a consistent mean and variance for each channel (Ioffe & Szegedy, 2015). This channel-wise normalization is preferable because the output features from multiple branches are concatenated with respect to channel dimension. These characteristics of batch normalization explain why it is still preferred for the decoder of segmentation networks, despite the existence of several alternatives, such as layer normalization, which does not perform channel-wise normalization (Ba et al., 2016). In summary, batch normalization provides a feature with a consistent scale, which allows the concatenation of several features from convolutional unit blocks.

**Bilinear Upsampling Breaks Scale Equilibrium**  However, even with batch normalization, feature scales exhibit disequilibrium when subsequently using bilinear upsampling. Consider a multi-level feature fusion for $\{\mathbf{P}_1, \mathbf{P}_2\}$, where each feature is an output of a convolutional unit block, and the latter $\mathbf{P}_2$ needs $r\times$ bilinear upsampling with $r > 1$ to become the same spatial size as $\mathbf{P}_1$. Fusing them requires computing

$$\mathbf{Z}^{\text{fuse}} = \mathbf{W}^{\text{fuse}}[\mathbf{P}_1; \text{UP}_r(\mathbf{P}_2)], \tag{7}$$

which is an intermediate result after convolutional layer of fusion with $\mathbf{W}^{\text{fuse}}$. Here, we investigate the scale of concatenation subjects. Although convolutional unit blocks on parallel branches assure consistent scales for $\{\mathbf{P}_1, \mathbf{P}_2\}$, scales of concatenation subjects $\{\mathbf{P}_1, \text{UP}_r(\mathbf{P}_2)\}$ are not guaranteed to be equal. Indeed, we claim that scale disequilibrium occurs during this feature fusion due to bilinear upsampling. Specifically, we prove that bilinear upsampling decreases feature variance:

**Theorem 3.2.** *Bilinear upsampling decreases feature variance, i.e.,* $\text{Var}[\text{UP}_r(\mathbf{X})] < \text{Var}[\mathbf{X}]$ *for upsampling ratio* $r > 1$ *and feature* $\mathbf{X}$ *that is not a constant feature.*

The constant feature here indicates a vector with the same constant elements. Note that bilinear upsampling conserves feature mean—but not feature variance. Furthermore, variance provides a suitably scaled result with respect to its size, which ensures that the decreased variance is not caused by the increased size due to upsampling. The decreased variance is caused by the linear interpolation function used in bilinear upsampling, which does not conserve the second moment that is included in the variance. See the Appendix A for a detailed proof and further discussion.

---

[1]For batch normalization $\gamma\hat{x} + \beta$, the initial state where $\gamma = 1$ and $\beta = 0$ provides a normalized feature $\hat{x}$.

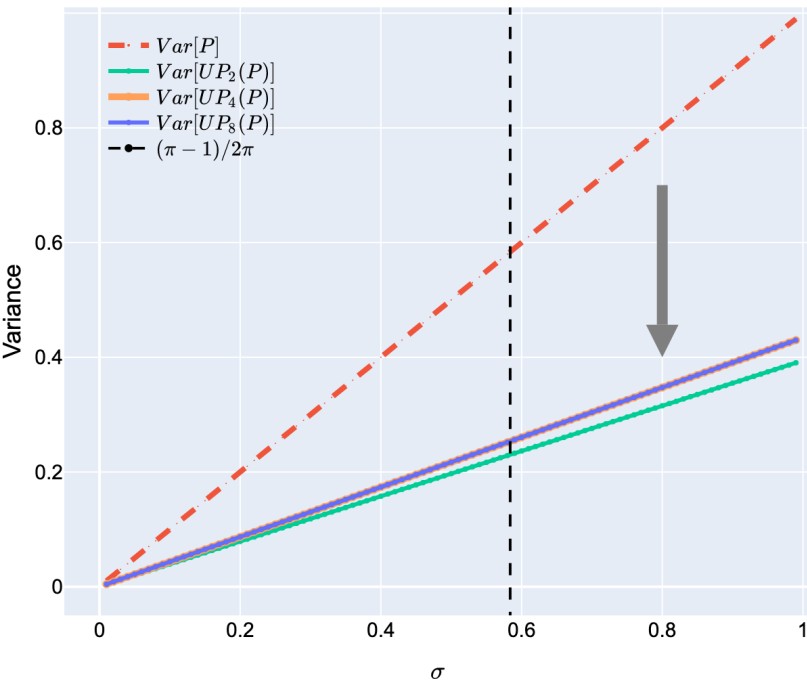

Figure 5: Empirical observation on decreased variance after bilinear upsampling. The black dotted line $(\pi - 1)/2\pi$ corresponds to the case when the output of a convolutional unit block is subjected to bilinear upsampling.

In Section 2, we reviewed multi-level feature fusion in modern decoders and found that, as a general rule, at least one branch uses the upsampling ratio $r_i = 1$, whereas others show $r_i > 1$. Therefore, Theorem 3.2 indicates that modern decoders with multi-level feature fusion exhibit scale disequilibrium. The fatal problem is that the last feature map always requires bilinear upsampling, which reduces its feature and gradient scales, obstructing the use of its rich information on an image. This problem arises even when using batch normalization: Because bilinear upsampling is applied after each convolutional unit block, the equalized scales subsequently change.

Note that bilinear upsampling is the de facto standard for semantic segmentation, and several studies have explicitly mentioned using bilinear upsampling in their papers (Zhao et al., 2017; Chen et al., 2017; Xiao et al., 2018). In consideration of this practice, our study targets upsampling with bilinear interpolation. Nevertheless, the decreased variance can also be observed for other interpolation methods such as bicubic, and our analysis and solution seamlessly apply to those upsampling methods.

**Empirical Observation**    Now, we empirically demonstrate decreased variance after bilinear upsampling. Considering a practical feature fusion scenario, we generated artificial random normal data $\mathbf{P}$ sampled from $\mathcal{N}(1/\sqrt{2\pi}, \sigma^2)$, which corresponds to a feature after a convolutional unit block but before bilinear upsampling. The feature $\mathbf{P}$ is set to have width 128, height 128, number of channels 256, and mini-batch size 16. Then we applied $r\times$ bilinear upsampling to $\mathbf{P}$ with $r \in \{2, 4, 8\}$ and measured the variance of each outcome. Figure 5 summarizes the results across various choices of $\sigma \in (0, 1)$. We observed that bilinear upsampling decreased feature variance in all simulations.

### 3.2 Proposed Solution: Scale Equalizer

Our claim is that we should modify the existing feature fusion method to achieve scale equilibrium for concatenation subjects—the output of each branch that ends with bilinear upsampling. This objective may be accomplished in several ways. The naive approach is to place batch normalization after bilinear upsampling,

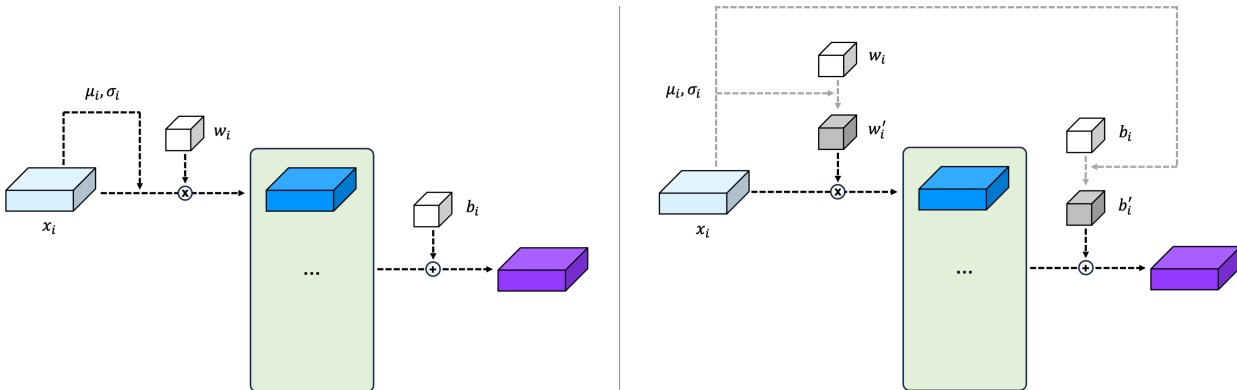

Figure 6: Scale equalizer with simple implementation (left) and efficient implementation (right). For efficient implementation, pre-computed global mean and std are applied to the weight and bias of the fusion layer in advance, as depicted by the gray dotted line. Here, the main training requires only black dotted lines, which maintains the same computational cost compared with the case without scale equalizers. Furthermore, we can remove the bias correction due to the subsequent batch normalization.

changing the pipeline from [Conv–BatchNorm–ReLU–UP] to [Conv–ReLU–UP–BatchNorm]. This pipeline yields a normalized feature with a consistent scale but requires extra computational cost. Because batch normalization computes the mean and standard deviation (std) of the current incoming feature map across the mini-batch, its computational complexity increases with the larger size of the feature (Huang et al., 2018). The computational complexity of the backward operation further increases with the larger size of the feature map because the derivative for batch normalization is much more complicated (Yao et al., 2021). Consequently, applying batch normalization to an upsampled feature causes a significantly more expensive computation compared with that of a non-upsampled feature. Considering this problem, we alternatively explore a computationally efficient solution to acquire scale equilibrium.

Here, we propose *scale equalizer*, a module to be injected after bilinear upsampling but before concatenation. To achieve scale equilibrium at minimal cost, we design the scale equalizer as simple as possible. Specifically, our proposed scale equalizer normalizes target feature $\mathbf{x}$ using global mean $\mu$ and global std $\sigma$ as $\mathrm{ScaleEqualizer}(\mathbf{x}) := (\mathbf{x} - \mu)/\sigma$. The global mean and std are scalars computed from the target feature $\mathbf{x}$ across the training dataset, which can be performed before training. Once the global mean and std are obtained, they can be set as fixed constants during training, which simplifies forward and backward operations for the scale equalizer. By contrast, mean and std are not constants for common normalization operations such as batch normalization or layer normalization because they use a mean and std of a current incoming feature. Thus, compared with existing normalization operations, the proposed scale equalizer can be implemented with little extra cost.

**Scale Equalizers Equalize Scales**   Now consider multi-level feature fusion with scale equalizers, where the scale equalizer is applied after bilinear upsampling of each branch but before concatenation. The concatenation subject $\mathrm{ScaleEqualizer}_i(\mathrm{UP}_{r_i}(\mathbf{P}_i))$ exhibits zero mean and unit variance, which assures scale equilibrium. Because the scale equalizer uses empirically measured values of the global mean and std, the scale equilibrium does not require architectural restrictions or specific conditions on weight. In other words, scale equilibrium is always guaranteed for any dataset and any architecture of segmentation network.

**Efficient Implementation via Initialization**   In fact, matching intra-layer gradient scales via scale equalizers can be interpreted as establishing a valid initialization. For multi-level feature fusion $y = \sum_i w_i x_i +$

---
**Algorithm 1** Efficient Implementation via Initialization

---
Input: set of training images $S$, encoder $e$, decoder $d$.
Using pretrained weights $\Theta$, initialize encoder $e$.
Using preferred initializers, initialize decoder $d$ into weight $\Omega$, including $\{w_i\}$.
Set $m_{1,i} = m_{2,i} = 0$ for $i \in \{1, \cdots, n_b\}$.
**for** $\mathbf{I} \in S$ **do**
    Extract FPN outputs $\{\mathbf{P}_i\}$ from $\mathbf{I}$ using encoder $e_\Theta$ and subnetwork of decoder $d_\Omega$.
    **for** $i = 1$ **to** $n_b$ **do**
        $m_{1,i} = m_{1,i} + \mathbb{E}[\mathrm{UP}_{r_i}(\mathbf{P}_i)]$.
        $m_{2,i} = m_{2,i} + \mathbb{E}[\mathrm{UP}_{r_i}(\mathbf{P}_i)^2]$.
    **end for**
**end for**
**for** $i = 1$ **to** $n_b$ **do**
    Obtain the global mean $\mu_i = m_{1,i}/|S|$.
    Obtain the global std $\sigma_i = \sqrt{m_{2,i}/|S| - \mu_i^2}$.
    Update $w_i$ in $\Omega$ using auxiliary initializer $w_i' = w_i/\sigma_i$.
**end for**
Using the updated decoder weight $\Omega'$, run the main training for encoder $e_\Theta$ and decoder $d_{\Omega'}$.

---

$b$, after replacing $x_i$ with $\mathrm{ScaleEqualizer}_i(x_i) = (x - \mu_i)/\sigma_i$, we obtain

$$y = \sum_i \left( \frac{w_i}{\sigma_i} \right) x_i + \left( b - \sum_i \frac{w_i \mu_i}{\sigma_i} \right). \tag{8}$$

Thus, injecting scale equalizers is equivalent to adopting an auxiliary initializer with $w_i' = w_i/\sigma_i$ and $b' = b - \sum_i w_i \mu_i/\sigma_i$. This auxiliary initializer means calibrating the weights and bias in the linear layer of fusion in advance using expected feature scales (Figure 6). Furthermore, because batch normalization follows subsequently (Section 2), the latter for bias correction is actually not needed, whereas the former for weight calibration is still needed to control the scales of concatenation subjects. For UPerHead as a concrete example, weights in the convolutional layer of fusion are partitioned into four groups with respect to channel dimension, and the weights in each group $w_i$ are re-scaled via the global std $\sigma_i$. In summary, after primary initialization of the decoder, we compute the global mean and std for each target feature, apply the auxiliary initializer to weights, and then proceed with the main training (Algorithm 1). This implementation requires no additional computational cost during main training, which enables us to achieve scale equilibrium for free.

As mentioned earlier, there may be other ways to achieve scale equilibrium by introducing complicated operations. Nevertheless, to demonstrate the effectiveness of scale equilibrium under the same computational cost, we opt for injecting scale equalizers and their efficient implementation through auxiliary initialization, which achieves scale equilibrium for free.

**Notes on Advantage of Scale Equalization** Even though the initial state exhibited scale disequilibrium for existing multi-level feature fusion, if neural network parameters are well optimized during training, it may achieve scale equilibrium after training. Nevertheless, our claim is that it would be better to achieve scale equilibrium from the initial state to avoid poor training dynamics. Additionally, note that the proposed scale equalizer does not introduce new learnable parameters in the neural network; rather, it behaves as constant scaling with fixed values. Therefore, even though scale equalizers are injected, the model's representation ability remains the same. While maintaining the same representation ability, the advantage of scale equalizers comes from the easier optimization in gradient descent by avoiding poor training dynamics (Section 3.1). This behavior would be rather similar to the initialization method of a neural network. For example, applying Xavier or He initialization (Glorot & Bengio, 2010; He et al., 2015) facilitates optimization in gradient descent, but they do not introduce new learnable parameters.

Table 2: Summarization of mIoU (%) from semantic segmentation experiments with multi-stage feature fusion using UPerHead. "Scale EQ" indicates the scale equalizers, and "Diff" indicates the mIoU difference after injecting the scale equalizers.

| Dataset | ADE20K | | | PASCAL VOC 2012 AUG | | |
|---|---|---|---|---|---|---|
| Encoder | w/o Scale EQ | w/ Scale EQ | Diff | w/o Scale EQ | w/ Scale EQ | Diff |
| Swin-T (Liu et al., 2021) | 43.384 | 43.576 | +0.192 | 78.750 | 78.996 | +0.246 |
| Swin-S | 47.298 | 47.486 | +0.188 | 81.940 | 82.138 | +0.198 |
| Swin-B | 47.490 | 47.648 | +0.158 | 82.200 | 82.378 | +0.178 |
| Twins-SVT-S (Chu et al., 2021) | 44.914 | 45.018 | +0.104 | 80.448 | 80.732 | +0.284 |
| Twins-SVT-B | 47.224 | 47.500 | +0.276 | 82.048 | 82.524 | +0.476 |
| Twins-SVT-L | 48.648 | 48.894 | +0.246 | 82.168 | 82.404 | +0.236 |
| ConvNeXt-T (Liu et al., 2022) | 45.024 | 45.300 | +0.276 | 80.668 | 80.932 | +0.264 |
| ConvNeXt-S | 47.736 | 47.866 | +0.130 | 82.472 | 82.650 | +0.178 |
| ConvNeXt-B | 48.376 | 48.684 | +0.308 | 82.934 | 83.038 | +0.104 |

## 4 Experiments

### 4.1 Multi-Stage Feature Fusion

**Objective** So far, we have discussed the need for scale equalizers for multi-level feature fusion. The objective here is to compare the segmentation performance before and after injecting scale equalizers into multi-stage feature fusion. We considered extensive setups, such as the choice of backbone and target dataset. For the backbone network, we employed recent vision transformers that achieved state-of-the-art performance. Nine backbones of Swin-{T, S, B} (Liu et al., 2021), Twins-SVT-{S, B, L} (Chu et al., 2021), and ConvNeXt-{T, S, B} (Liu et al., 2022) pretrained on ImageNet-1K (Deng et al., 2009) were examined, where T, S, B, and L stand for tiny, small, base, and large models, respectively. These encoders require bilinear upsampling for multi-stage feature fusion. Targeting multi-stage feature fusion, we employed UPerHead (Xiao et al., 2018). Two datasets were examined, including the ADE20K (Zhou et al., 2019) and PASCAL VOC 2012 (Everingham et al., 2015).

**Hyperparameters** To follow common practice for semantic segmentation, training recipes from `MMSegmentation` (Contributors, 2020) were employed. For training with Swin and Twins encoders, AdamW optimizer (Loshchilov & Hutter, 2019) with weight decay $10^{-2}$, betas $\beta_1 = 0.9, \beta_2 = 0.999$, and learning rate $6 \times 10^{-5}$ with polynomial decay of the 160K scheduler after linear warmup were used. For training with ConvNeXt encoders, AdamW optimizer with weight decay $5 \times 10^{-2}$, betas $\beta_1 = 0.9, \beta_2 = 0.999$, learning rate $10^{-4}$ with polynomial decay of the 160K scheduler after linear warmup, and mixed precision training (Micikevicius et al., 2018) were used. The training was conducted on a $4 \times$ GPU machine, and SyncBN (Zhang et al., 2018) was used for distributed training. We measured the mean intersection over union (mIoU) and reported the average of five runs with different random seeds.

**Datasets** The ADE20K dataset contains scene-centric images along with the corresponding segmentation labels. A crop size of $512 \times 512$ pixels was used, which was obtained after applying mean-std normalization and a random resize operation using a size of $2048 \times 512$ pixels with a ratio range of 0.5 to 2.0. Furthermore, a random flipping with a probability of 0.5 and the photometric distortions were applied. The objective was to classify each pixel into one of the 150 categories and train the segmentation network using the pixel-wise cross-entropy loss. The same goes for the PASCAL VOC 2012 dataset with 21 categories, and we followed the augmented PASCAL VOC 2012 dataset.

**Results** We observed that injecting scale equalizers into multi-stage feature fusion improved the mIoU index compared with the same models without scale equalization (Table 2). The mIoU increases of about +0.1 to +0.4 were consistently observed across all setups of nine backbones and two datasets. Note that the scale disequilibrium arises within the decoder at the concatenation layer after upsampling. Therefore,

Table 3: Summarization of mIoU (%) from semantic segmentation experiments with single-stage feature fusion using various heads.

| Dataset | Cityscapes | | | ADE20K | | |
|---|---|---|---|---|---|---|
| Decoder | w/o Scale EQ | w/ Scale EQ | Diff | w/o Scale EQ | w/ Scale EQ | Diff |
| FCNHead (Long et al., 2015) | 76.578 | 76.972 | +0.394 | 39.780 | 39.958 | +0.178 |
| PSPHead (Zhao et al., 2017) | 79.394 | 79.858 | +0.464 | 43.970 | 44.228 | +0.258 |
| ASPPHead (Chen et al., 2017) | 79.312 | 79.720 | +0.408 | 44.854 | 45.004 | +0.150 |
| SepASPPHead (Chen et al., 2018b) | 80.448 | 80.592 | +0.144 | 45.144 | 45.486 | +0.342 |

this problem is related to the architectural design of the decoder. Using a larger encoder network, albeit having much expressivity, cannot solve this problem. This explains why scale equalization matters from tiny to large backbone models. Furthermore, our method goes beyond the trade-off between computational cost and performance. The proposed method does not introduce additional layers; while keeping the same architecture and expressive power of the deep neural network, the performance gain of the proposed method can be obtained in actually free (Section 3.2). In other words, scale equalization provides a free performance gain without incurring additional computational expenses.

## 4.2 Single-Stage Feature Fusion

**Objective** Now we examine single-stage feature fusion. The target encoder was ResNet-101 (He et al., 2016), which was modified to exhibit output stride $s = 8$ and was pretrained on ImageNet-1K. We targeted most standard and popular decoders, including FCNHead (Long et al., 2015), PSPHead (Zhao et al., 2017), ASPPHead (Chen et al., 2017), and SepASPPHead (Chen et al., 2018b). The target datasets were the Cityscapes (Cordts et al., 2016) and ADE20K datasets.

**Hyperparameters** Similar to Section 4.1, training recipes from `MMSegmentation` were employed. For training on the Cityscapes dataset, stochastic gradient descent with momentum 0.9, weight decay $5 \times 10^{-4}$, and learning rate $10^{-2}$ with polynomial decay of the 80K scheduler were used. The same goes for training on the ADE20K dataset while using the 160K scheduler instead. The training was conducted on a $4\times$ GPU machine, and SyncBN was used for distributed training. We measured the mIoU and reported the average of five runs with different random seeds.

**Datasets** The Cityscapes dataset contains images of urban street scenes along with the corresponding segmentation labels. A crop size of $1024 \times 512$ pixels was used, which was obtained after applying mean-std normalization and a random resize operation using a size of $2048 \times 1024$ pixels with a ratio range of 0.5 to 2.0. Furthermore, a random flipping with a probability of 0.5 and the photometric distortions were applied. The objective was to classify each pixel into one of the 19 categories and train the segmentation network using the pixel-wise cross-entropy loss. Experiments on the ADE20K followed the description in Section 4.1.

**Results** Similarly, injecting scale equalizers consistently improved the mIoU index across all setups of four decoder heads and two datasets (Table 3), which verifies the effectiveness of scale equalizers for any choice of architecture. See the Appendix D for more experimental results.

## 5 Discussion

### 5.1 Comparison With Other Normalization Layers

Consider the pipeline used for generating each concatenation subject in multi-level feature fusion. The existing pipeline of [Conv–BatchNorm–ReLU–UP] yields concatenation subjects with different variances due to the last upsampling. Here, when modifying it into [Conv–ReLU–UP–BatchNorm], it outputs a normalized

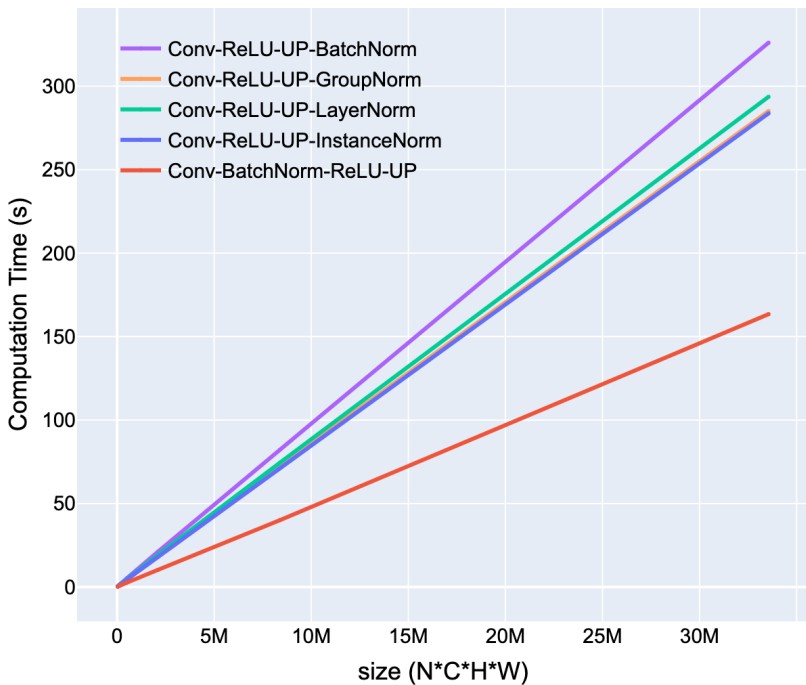

Figure 7: When modifying the existing pipeline from [Conv–BatchNorm–ReLU–UP] into others such as [Conv–BatchNorm–ReLU–UP], computation time significantly increases.

Table 4: Experimental results on different pipelines for the ADE20K dataset using Swin-T.

| Pipeline | mIoU (%) |
|---|---|
| Conv–BatchNorm–ReLU–UP | 43.384 |
| Conv–BatchNorm–ReLU–UP–ScaleEQ | 43.576 |
| Conv–ReLU–UP–BatchNorm | 39.998 |
| Conv–ReLU–UP–GroupNorm | 43.212 |
| Conv–ReLU–UP–LayerNorm | 42.870 |

feature, which achieves an equalized scale across concatenation subjects. This behavior can also be achieved with other normalization layers, such as GroupNorm and LayerNorm.

Although the use of a normalization layer after upsampling can be another solution to achieve scale equilibrium, it causes increased computational costs due to the enlarged size of the feature map. Because mean and std are computed for the incoming feature map, the larger size of the feature map requires much computational cost for computing the mean and std. Furthermore, the computed mean and std are used for normalization of each element, which leads to increased complexity overall.

This behavior can be verified through simulation. Using $\mathbf{x} \in \mathbb{R}^{N \times C \times H \times W}$ for $N = 16$ and $C = 128$, we simulated $8\times$ bilinear upsampling and compared the computation time required for the four pipelines: [Conv–BatchNorm–ReLU–UP], [Conv–ReLU–UP–BatchNorm], [Conv–ReLU–UP–GroupNorm], and [Conv–ReLU–UP–LayerNorm]. Figure 7 summarizes the results. We observed that the existing [Conv–BatchNorm–ReLU–UP] pipeline consistently exhibited faster computation, whereas modified pipelines showed significantly slower computation. Note that the computational cost in case of injecting scale equalizers is equal to the existing pipeline of [Conv–BatchNorm–ReLU–UP] because it can be implemented without extra cost

Table 5: Experimental results on the monocular depth estimation task.

| Model | $\delta < 1.25 \uparrow$ | $\delta < 1.25^2 \uparrow$ | $\delta < 1.25^3 \uparrow$ | Abs Rel $\downarrow$ | RMSE $\downarrow$ | log10 $\downarrow$ | RMSE log $\downarrow$ | SILog $\downarrow$ | Sq Rel $\downarrow$ |
|---|---|---|---|---|---|---|---|---|---|
| GEDepth-Vanilla | 0.9763 | 0.9971 | 0.9994 | 0.0498 | 2.0416 | 0.0218 | 0.0766 | 7.0109 | 0.1429 |
| GEDepth-Vanilla w/ Scale EQ | 0.9768 | 0.9972 | 0.9994 | 0.0492 | 2.0180 | 0.0215 | 0.0760 | 6.9702 | 0.1400 |
| GEDepth-Adaptive | 0.9751 | 0.9970 | 0.9993 | 0.0495 | 2.0909 | 0.0218 | 0.0776 | 7.1189 | 0.1479 |
| GEDepth-Adaptive w/ Scale EQ | 0.9757 | 0.9971 | 0.9993 | 0.0494 | 2.0683 | 0.0217 | 0.0771 | 7.0763 | 0.1457 |

(Section 3.2, Algorithm 1); therefore, in terms of computational complexity, we claim that our proposed method is superior compared with the use of a normalization layer.

In addition to computational costs, we compared segmentation performance when using modified pipelines (Table 4). We empirically observed that modifying the existing ordering of [Conv–BatchNorm–ReLU–UP] has a side effect of degraded segmentation performance, which outweighs possible advantages. This phenomenon was consistently confirmed for BatchNorm, GroupNorm, and LayerNorm. Although GroupNorm or LayerNorm might yield improved performance compared with BatchNorm, their performances were even below the baseline performance of the existing pipeline of [Conv–BatchNorm–ReLU–UP]. Furthermore, applying a normalization layer after an upsampled feature requires much computational cost. Considering both computational cost and segmentation performance, the best pipeline is our proposed pipeline of [Conv–BatchNorm–ReLU–UP–ScaleEQ]. In consideration of this, we opt for keeping the existing pipeline without modification in its ordering while injecting a scale equalizer at the end of the pipeline.

### 5.2 Scale Equalization for Other Tasks

We find that the use of upsampling and multi-level feature fusion is prevalent in the machine learning community. Although we focused on the multi-level feature fusion in semantic segmentation networks as a prime example, our scale equalization matters for other multi-level feature fusion tasks. Specifically, scale equalization generally matters for modern encoder-decoder networks. For example, monocular depth estimation networks have used the encoder-decoder architecture, whose multi-level feature fusion exhibits scale disequilibrium similar to the semantic segmentation networks.

In consideration of this, we additionally verified scale equalization for a monocular depth estimation network. The target model was GEDepth (Yang et al., 2023), where its feature fusion module concatenates upsampled and non-upsampled feature maps. Using the KITTI dataset (Geiger et al., 2013), we trained the model with and without scale equalizers in the feature fusion module (Table 5). We observed that injecting scale equalizers improved the performance of the monocular depth estimation task, which connotes that scale equalization similarly matters for other tasks where encoder-decoder architecture is deployed.

### 5.3 Qualitative Analysis

Figure 8 provides segmentation examples for the ADE20K dataset. We find that injecting scale equalizers leads to a better understanding of the global context of images. Specifically, with scale equalizers, the global layout is better captured for large parts. Indeed, our analysis says that existing multi-level feature fusion suffers from scale disequilibrium, which lowers the contribution of the last feature map that contains rich global information. Here, injecting scale equalizers facilitates the last feature map to be involved in multi-level feature fusion, which leads to a better understanding of the global context of the image.

## 6 Conclusion

This study discussed the scale disequilibrium in multi-level feature fusion for semantic segmentation tasks. First, we reviewed the mechanisms of existing segmentation networks, which perform multi- or single-stage feature fusion. We demonstrated that the existing multi-level feature fusion exhibits scale disequilibrium due to bilinear upsampling, which causes degraded gradient descent optimization. To address this problem, injecting scale equalizers is proposed to guarantee scale equilibrium for multi-level feature fusion. Experi-

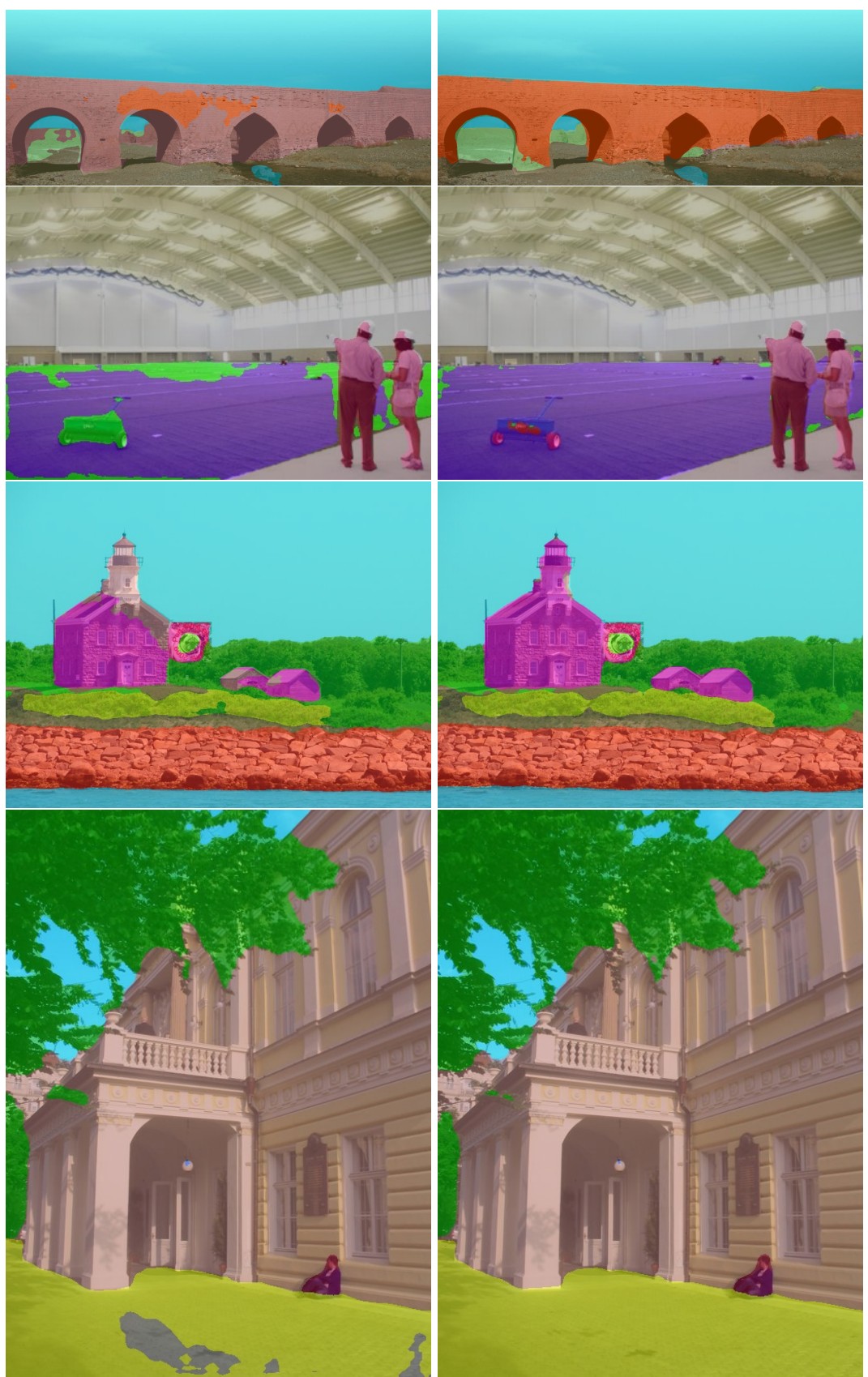

Figure 8: Segmentation examples without (Left) and with (Right) scale equalizers.

ments showed that the use of scale equalizers consistently increased the mIoU index by about +0.1 to +0.4 across numerous datasets and architectures. We hope that our proposed problem and solution will facilitate the research community in developing an improved multi-level feature fusion and segmentation network.

## Acknowledgments

This work was supported by Samsung Electronics Co., Ltd (IO201210-08019-01).

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

# A    Proof of Theorem 3.2

**Notation**   In this section, we prove Theorem 3.2. Here, we consider 1D bilinear upsampling, because 2D bilinear sampling is a straightforward extension of it. For a given sequence $\mathbf{X} = \{X_1, \cdots, X_n\}$ of size $n$, applying $r\times$ bilinear upsampling yields a sequence $\mathrm{UP}_r(\mathbf{X})$ of size $rn$. Because these sequences are discrete, bilinear upsampling can be thought of as a coordinate transformation, which transforms coordinate $\{p_1, \cdots, p_n\}$ into coordinate $\{q_1, \cdots, q_{rn}\}$. We follow the common option of bilinear upsampling to set `align_corners=False`, where each coordinate is regularly spaced and shares its center. Within $p_i$ and $p_{i+1}$ for $i \in \{1, \cdots, n-1\}$, the upsampled coordinate has $r$ regularly spaced points as $q_{r(i-0.5)+j} = p_i + (2j-1)l$ for $l = \frac{p_{i+1}-p_i}{2r}$ and $j \in \{1, \cdots, r\}$. Because bilinear upsampling is a piece-wise linear interpolation, we represent the piece-wise linear function as $f(x)$ where 1) for coordinate point $p_i$, we define $f(p_i) := X_i$ for $i \in \{1, \cdots, n\}$, 2) following the behavior of `align_corners=False`, we define $f(x) := X_1$ on left outer interval $x \in (-\infty, p_1)$ and $f(x) := X_n$ on right outer interval $x \in (p_n, +\infty)$, and 3) on $i$th interval $x \in (p_i, p_{i+1})$, we define $f(x)$ as a linear line $f(x) := a_i x + b_i$ connecting two points $(p_i, X_i)$ and $(p_{i+1}, X_{i+1})$ where $a_i = \frac{X_{i+1}-X_i}{p_{i+1}-p_i}$ and $b_i = \frac{X_i p_{i+1} - X_{i+1} p_i}{p_{i+1}-p_i}$. Using this notation, we represent bilinear upsampling as a transformation from the original data $\{f(p_1), \cdots, f(p_n)\}$ to its upsampled data $\{f(q_1), \cdots, f(q_{rn})\}$. Finally, we represent the mean of a sequence by sampling $f$ as $\mathbb{E}_{x\sim p}[f(x)] = \frac{1}{n} \sum_{i=1}^{n} f(p_i)$ and the same goes for variance.

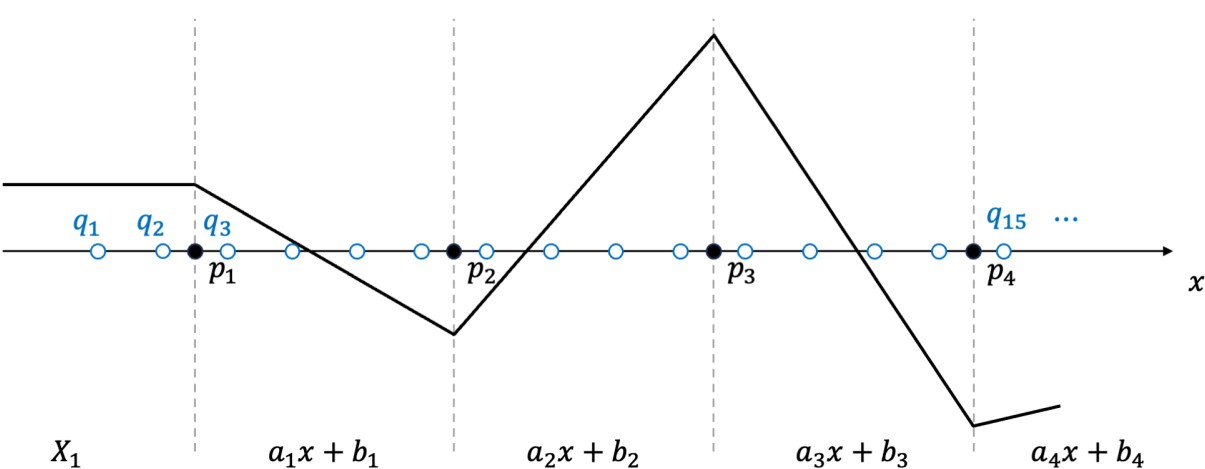

Figure 9: Example of two coordinates and a piece-wise linear function for $4\times$ bilinear upsampling.

**Objective**   Firstly, we know that bilinear upsampling conserves feature mean, *i.e.*, $\mathbb{E}[\mathrm{UP}_r(\mathbf{X})] = \mathbb{E}[\mathbf{X}]$. Using the above-mentioned notation, this property can be represented as $\mathbb{E}_{x\sim p}[f(x)] = \mathbb{E}_{x\sim q}[f(x)]$. In fact, mean-conservation holds by the definition of the two coordinates because they share the center. Now, we investigate feature variance before and after bilinear upsampling. Because $\mathrm{Var}[X] = \mathbb{E}[X^2] - (\mathbb{E}[X])^2$, we inspect the second moment $\mathbb{E}[X^2]$ for the two coordinates. In other words, we compare $\mathbb{E}_{x\sim p}[(f(x))^2]$ and $\mathbb{E}_{x\sim q}[(f(x))^2]$.

**Main Proof**   Firstly, we write

$$\mathbb{E}_{x\sim p}[(f(x))^2] = \frac{1}{n} \sum_{i=1}^{n} (f(p_i))^2 \tag{9}$$

$$= \frac{1}{2n}[(f(p_1))^2 + \{(f(p_1))^2 + (f(p_2))^2\} + \cdots + \{(f(p_{n-1}))^2 + (f(p_n))^2\} + (f(p_n))^2], \tag{10}$$

$$\mathbb{E}_{x\sim q}[(f(x))^2] = \frac{1}{rn}[(f(q_1))^2 + \cdots + (f(q_{rn}))^2]. \tag{11}$$

This expression enables us to compare sub-terms of the above for two coordinates on each interval. For $i \in \{1, \cdots, n-1\}$, we define their sub-terms as

$$P_i := \frac{(f(p_i))^2 + (f(p_{i+1}))^2}{2n}, \tag{12}$$

$$Q_i := \frac{(f(q_{r(i-0.5)+1}))^2 + \cdots + (f(q_{r(i+0.5)}))^2}{rn}. \tag{13}$$

Now we investigate $P_i - Q_i$. Note that $\frac{1}{2n}(f(p_i))^2 = (\frac{r}{2})(\frac{1}{rn})(f(p_i))^2$, which can be interpreted as repeating $\frac{1}{rn}(f(p_i))^2$ in $\frac{r}{2}$ times. Using this representation, we obtain

$$rn(P_i - Q_i) = \{\underbrace{(f(p_i))^2 + \cdots + (f(p_i))^2}_{r/2 \text{ terms}}\} + \{\underbrace{(f(p_{i+1}))^2 + \cdots + (f(p_{i+1}))^2}_{r/2 \text{ terms}}\} \tag{14}$$

$$- \{\underbrace{(f(q_{r(i-0.5)+1}))^2 + \cdots + f(q_{ri})^2}_{r/2 \text{ terms}}\} - \{\underbrace{(f(q_{ri+1}))^2 \cdots (f(q_{r(i+0.5)})^2)}_{r/2 \text{ terms}}\} \tag{15}$$

$$= \{(f(p_i))^2 - (f(q_{r(i-0.5)+1}))^2\} + \cdots + \{(f(p_i))^2 - (f(q_{ri}))^2\}$$
$$+ \{(f(p_{i+1}))^2 - (f(q_{ri+1}))^2\} + \cdots + \{(f(p_{i+1}))^2 - (f(q_{r(i+0.5)}))^2\}. \tag{16}$$

Thus, the first $r/2$ terms indicate the squared difference between two coordinates on the left half area, whereas the last $r/2$ terms similarly come from the two coordinates on the right half area.

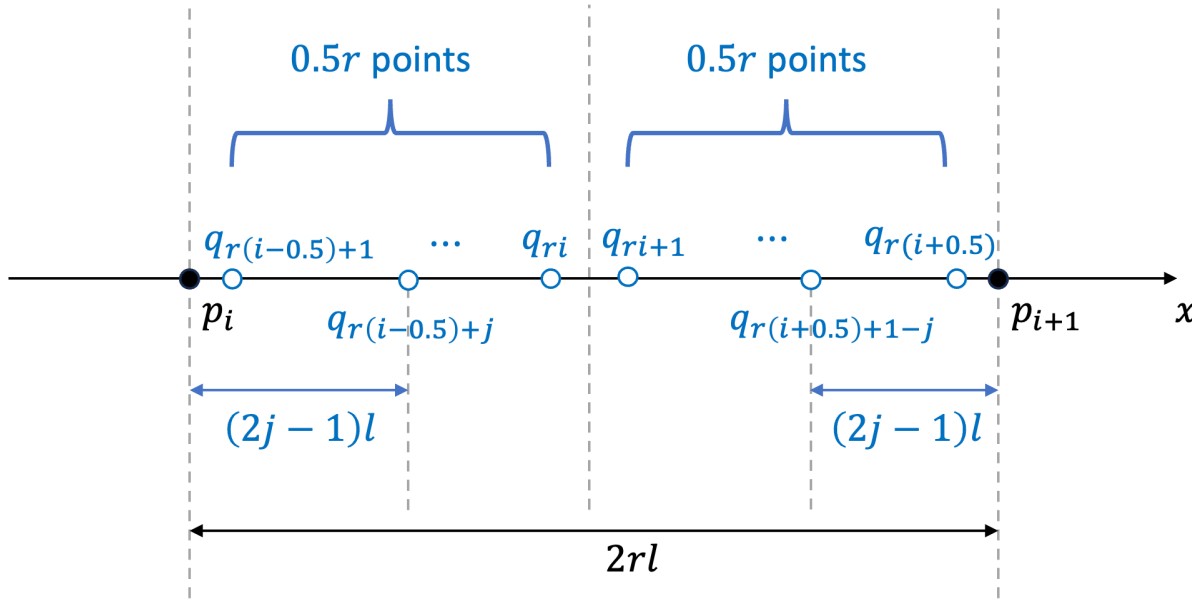

Figure 10: Illustration of two coordinates on the $i$th interval of $x \in (p_i, p_{i+1})$.

For $q_{r(i-0.5)+j}$ on the left half area with $j \in \{1, \cdots, r/2\}$, we have $p_i - q_{r(i-0.5)+j} = -(2j-1)l$. Because $f(x)$ is defined as $f(x) = a_i x + b_i$ on the $i$th interval $x \in (p_i, p_{i+1})$, we know that $f(x_1) - f(x_2) = a_i(x_1 - x_2)$ and $f(x_1) + f(x_2) = a_i(x_1 + x_2) + 2b_i$. Thus, we have

$$(f(p_i))^2 - (f(q_{r(i-0.5)+j}))^2 = \{f(p_i) - f(q_{r(i-0.5)+j})\}\{f(p_i) + f(q_{r(i-0.5)+j})\} \tag{17}$$

$$= a_i(p_i - q_{r(i-0.5)+j})\{a_i(p_i + q_{r(i-0.5)+j}) + 2b_i\} \tag{18}$$

$$= -a_i l(2j-1)\{a_i(p_i + q_{r(i-0.5)+j}) + 2b_i\}. \tag{19}$$

Similarly, on the right half area, we consider $q_{r(i+0.5)+1-j}$ with $j \in \{1, \cdots, r/2\}$, which is in reversed order. We have $p_i - q_{r(i+0.5)+1-j} = (2j-1)l$, and the right half area shares $f(x) = a_i x + b_i$ on the left half area. Therefore, we obtain

$$(f(p_i))^2 - (f(q_{r(i+0.5)+1-j}))^2 = \{f(p_i) - f(q_{r(i+0.5)+1-j})\}\{f(p_i) + f(q_{r(i+0.5)+1-j})\} \tag{20}$$

$$= a_i(p_i - q_{r(i+0.5)+1-j})\{a_i(p_i + q_{r(i+0.5)+1-j}) + 2b_i\} \tag{21}$$

$$= a_i l(2j-1)\{a_i(p_i + q_{r(i+0.5)+1-j}) + 2b_i\}. \tag{22}$$

Sum of Eqs. 19 and 22 yields

$$\{(f(p_i))^2 - (f(q_{r(i-0.5)+j}))^2\} + \{(f(p_i))^2 - (f(q_{r(i+0.5)+1-j}))^2\}$$
$$= a_i^2 l(2j-1)\{(p_{i+1} - p_i) + (q_{r(i+0.5)+1-j} - q_{r(i-0.5)+j})\} \geq 0. \tag{23}$$

Equality holds if $a_i = 0$, which requires $X_{i+1} = X_i$. In summary, Eq. 16 can be written as the sum of $j$th terms for $j \in \{1, \cdots, r/2\}$, where each $j$th term is always non-negative. Thus, we conclude that $P_i - Q_i \geq 0$ for $i \in \{1, \cdots, n-1\}$.

So far, we have compared the sub-terms of Eqs. 10 and 11 for two coordinates on each of the $i$th interval for $i \in \{1, \cdots, n-1\}$. We are left with two sub-terms on the left outer interval and the right outer interval. Fortunately, by the definition of bilinear upsampling on outer intervals, we have

$$\frac{1}{2n}[(f(p_1))^2 + (f(p_n))^2] = \frac{1}{rn}[\underbrace{\{(f(p_1))^2 + \cdots + (f(p_1))^2\}}_{r/2 \text{ terms}} + \underbrace{\{(f(p_n))^2 + \cdots + (f(p_n))^2\}}_{r/2 \text{ terms}}] \tag{24}$$

$$= \frac{1}{rn}[\{(f(q_1))^2 + \cdots + (f(q_{0.5r}))^2\} + \{(f(q_{r(n-0.5)+1}))^2 + \cdots + (f(q_{rn}))^2\}]. \tag{25}$$

Thus, on the outer intervals, sub-terms of Eqs. 10 and 11 are identical.

Finally, from Eqs. 10, 11, 23, and 25, we obtain $\mathbb{E}_{x \sim p}[(f(x))^2] \geq \mathbb{E}_{x \sim q}[(f(x))^2]$. Note that equality holds if $a_i = 0$, i.e., $X_{i+1} = X_i$ for all $i \in \{1, \cdots, n-1\}$, which is only satisfied if the original feature is a constant feature. However, because we consider original data that is not a sequence of constants, we know that at least one of $a_i$ exhibits $a_i \neq 0$, which yields $\mathbb{E}_{x \sim p}[(f(x))^2] > \mathbb{E}_{x \sim q}[(f(x))^2]$. This inequality leads to $\text{Var}_{x \sim p}[f(x)] > \text{Var}_{x \sim q}[f(x)]$. Therefore, we conclude that applying bilinear upsampling decreases variance.

## B  Mean and Variance of Convolutional Unit Block

In the main text, we wrote

$$\mathbb{E}[\text{ReLU}(\text{BatchNorm}(\mathbf{W}\mathbf{x}))] = \frac{1}{\sqrt{2\pi}}, \tag{26}$$

$$\text{Var}[\text{ReLU}(\text{BatchNorm}(\mathbf{W}\mathbf{x}))] = \frac{\pi - 1}{2\pi}. \tag{27}$$

Here, we derive these properties. Because we consider a decoder in an initialized state where $\gamma = 1$ and $\beta = 0$, batch normalization provides a normalized feature $z \sim \mathcal{N}(0, 1)$, where its probability density function is $p(z) = \frac{1}{\sqrt{2\pi}} \exp\left(-\frac{x^2}{2}\right)$. Hence, we examine the mean and variance after ReLU, i.e., $\mathbb{E}[\text{ReLU}(z)]$ and $\text{Var}[\text{ReLU}(z)]$. In the case of the mean, we have

$$\mathbb{E}[\text{ReLU}(z)] = \int_{-\infty}^{\infty} \text{ReLU}(z)p(z)dz \tag{28}$$

$$= \int_{0}^{\infty} zp(z)dz \tag{29}$$

$$= \frac{1}{\sqrt{2\pi}}, \tag{30}$$

where the last equation holds owing to the properties of a half-normal or truncated normal distribution. In the case of the second moment, we derive

$$\mathbb{E}[(\text{ReLU}(z))^2] = \int_{-\infty}^{\infty} (\text{ReLU}(z))^2 p(z) dz \tag{31}$$

$$= \int_{0}^{\infty} z^2 p(z) dz \tag{32}$$

$$= \frac{1}{2} \int_{-\infty}^{\infty} z^2 p(z) dz \tag{33}$$

$$= \frac{1}{2} \mathbb{E}[z^2] \tag{34}$$

$$= \frac{1}{2}, \tag{35}$$

where the third equation holds by even symmetry. Thus, we obtain

$$\text{Var}[\text{ReLU}(z)] = \mathbb{E}[(\text{ReLU}(z))^2] - (\mathbb{E}[\text{ReLU}(z)])^2 \tag{36}$$

$$= \frac{1}{2} - \frac{1}{2\pi} \tag{37}$$

$$= \frac{\pi - 1}{2\pi}, \tag{38}$$

which concludes the proof.

## C   Python Code for Fig. 5

```python
import torch
import torch.nn.functional as F
import math

N, H, W, C = 16, 128, 128, 256

for v in range(1, 100):
    v = v / 100.0
    mean = 0.3989 * torch.ones(N, C, H, W)
    std = math.sqrt(v)
    x = torch.normal(mean=mean, std=std).cuda()

    A_1 = x
    print(A_1.var(unbiased=False).item())
    del A_1

    A_2 = F.interpolate(x, size=(2 * H, 2 * W), mode="bilinear", align_corners=False)
    print(A_2.var(unbiased=False).item())
    del A_2

    A_4 = F.interpolate(x, size=(4 * H, 4 * W), mode="bilinear", align_corners=False)
    print(A_4.var(unbiased=False).item())
    del A_4

    A_8 = F.interpolate(x, size=(8 * H, 8 * W), mode="bilinear", align_corners=False)
    print(A_8.var(unbiased=False).item())
    del A_8

    print("")
    del x
```

Listing 1: Python code for Fig. 5

# D  Additional Experimental Results

We additionally provide experimental results on single-stage feature fusion using ResNet-50.

Table 6: Summarization of mIoU (%) from semantic segmentation experiments with single-stage feature fusion using various heads.

| Dataset | ADE20K | | |
|---|---|---|---|
| Decoder | w/o Scale EQ | w/ Scale EQ | Diff |
| FCNHead | 36.544 | 36.998 | +0.454 |
| PSPHead | 41.666 | 41.806 | +0.140 |
| ASPPHead | 42.934 | 43.220 | +0.286 |
| SepASPPHead | 43.910 | 44.056 | +0.146 |

