# OpenReview forum: "Scale Equalization for Multi-Level Feature Fusion"
_TMLR — Accepted by TMLR_

### Review · Reviewer_7Qfk · 2024-05-27

**Summary Of Contributions:**

This study discussed the scale disequilibrium in multi-level feature fusion. We demonstrated that the existing multi-level feature fusion exhibits scale disequilibrium due to bilinear upsampling, which causes degraded gradient descent optimization. To address this problem, injecting scale equalizers is proposed to guarantee scale equilibrium for multi-level feature fusion. Experiments showed that the use of scale equalizers consistently increased the mIoU index by about +0.1 to +0.4 across numerous datasets and architectures.

**Audience:**

No

**Claims And Evidence:**

Yes

**Requested Changes:**

No

**Strengths And Weaknesses:**

Questions:
The author highlights that combining features of varying scales can cause a scale disequilibrium issue within the model, a pervasive and undesirable defect that adversely affects gradient descent. Could you conduct additional experiments on other computer vision tasks? For instance, exploring detection tasks, classification tasks, and temporal forecasting tasks (such as weather forecasting), which also depend on multi-level feature fusion, could provide further insights.
Given that neural network parameters are updated during model training, do you believe that a well trained model parameter could potentially mitigate or even resolve this disequilibrium issue?
Do you think these kind of disequilibrium could increase the mode’s representation ability?
The author highlight bilinear upsampling will cause degraded gradient descent optimization. Will nearest-neighbor interpolation, transpose Conv suffer from the same problem or not?

---

> ### Author Response · Authors · 2024-08-07
> **Response to Reviewer 7Qfk**
>
> Thank you for thoughtful comments to improve the quality of this manuscript. We find that your comments are valid; in our revised manuscript, we reflected them as much as possible. Here, we provide a point-to-point response to your requested changes.
>
> > The author highlights that combining features of varying scales can cause a scale disequilibrium issue within the model, a pervasive and undesirable defect that adversely affects gradient descent. Could you conduct additional experiments on other computer vision tasks? For instance, exploring detection tasks, classification tasks, and temporal forecasting tasks (such as weather forecasting), which also depend on multi-level feature fusion, could provide further insights.
>
> Thank you for your valuable comment. First, we find that the use of upsampling and multi-level feature fusion is quite prevalent in the machine learning community. Although we focused on the multi-level feature fusion in semantic segmentation networks as a prime example, our scale equalization would matter for other multi-level feature fusion tasks. Specifically, our scale equalizers generally matter for modern encoder-decoder networks. For example, monocular depth estimation networks have used the encoder-decoder architecture, whose multi-level feature fusion exhibits scale disequilibrium similar to the semantic segmentation networks.
>
> In consideration of this, we additionally verified scale equlization for a monocular depth estimation network. The target model was GEDepth, where its feature fusion module concatenates upsampled and non-upsampled feature maps. We compared performance before and after injecting scale equalizers into the feature fusion module (Table 5). We observed that injecting scale equalizers improved the performance of the monocular depth estimation task.
>
> In summary, we believe that scale equalization matters for other tasks where encoder-decoder architecture is deployed. In the revised manuscript, to support the general applicability of the scale equalizer, we mentioned this point by providing the additional experiment on monocular depth estimation.
>
> > - Given that neural network parameters are updated during model training, do you believe that a well trained model parameter could potentially mitigate or even resolve this disequilibrium issue?
> > - Do you think these kind of disequilibrium could increase the mode’s representation ability?
>
> Because these two points are connected, we answer them together. Indeed, this is an interesting topic. Note that the proposed scale equalizer does not introduce new learnable parameters in the neural network; rather, it behaves as constant scaling with fixed values. Therefore, even though scale equalizers are injected, the model’s representation ability remains the same. While maintaining the same representation ability, the advantage of scale equalizers comes from the easier optimization in gradient descent (Section 3.1). This behavior would be rather similar to the initialization method of a neural network. For example, applying Xavier or He initialization facilitates optimization in gradident descent, but they do not introduce new learnable parameters. Indeed, even though the initial state exhibited scale disequilibrium for existing multi-level feature fusion, if neural network parameters are well optimized during training, it may exhibit scale equilibrium after training. Nevertheless, our claim is that it would be better to achieve scale equilibrium from the initial state to remove difficulty in early training.
>
> After all, thank you for pointing out this issue. We believe that this analysis is worthy of being mentioned. In our revised manuscript, we mentioned this issue to clarify the advantage of injecting a scale equalizer.

---

> ### Author Response · Authors · 2024-08-07
> **Response to Reviewer 7Qfk**
>
> > The author highlight bilinear upsampling will cause degraded gradient descent optimization. Will nearest-neighbor interpolation, transpose Conv suffer from the same problem or not?
>
> Thank you for your insightful comments. Firstly, because the transpose Conv can be regarded as introducing the nearest neighbor during operation, here we will discuss the nearest neighbor and bilinear upsampling.
>
> As you said, our study focuses on bilinear upsampling. Indeed, the “resize” operation in several libraries acts as bilinear upsampling by default. Furthermore, bilinear upsampling is the de facto standard for semantic segmentation. For example, the authors of PSPNet said, “… we resize them via bilinear interpolation to the size of …” in their paper. Others such as UPerNet and DeepLab also explicitly mentioned using bilinear upsampling in their papers. In consideration of these practices, we targeted bilinear upsampling for our topic. We observed that other fields seldomly use bicubic upsampling, such as when upsampling the positional embedding in vision transformers, but we did not observe using other upsampling.
>
> Indeed, we agree that only repeatition-style upsampling would conserve both the mean and variance, whereas all others, including bilinear and bicubic upsampling, would not. This coincides with your comment. Indeed, nearest neighbor upsampling corresponds to repeatition-style upsampling and conserves both the mean and variance of a feature because nearest neighbor upsampling can be interpreted as duplicating features and shuffling as continuous order, which conserves both the mean and variance. However, we found that when training the segmentation network with the nearest neighbor, significantly degraded mIoU results were observed. We conjecture that the semantic segmentation task requires a more natural upsampling method, such as bilinear or bicubic, but the nearest neighbor provides too simple and unnatural upsampling. In other words, though the nearest neighbor conserves feature scale, it has the side effect of degraded upsampling. Therefore, in semantic segmentation, we should use bilinear or (possibly) bicubic upsampling to obtain natural upsampling. However, because they do not conserve variance, we should explicitly apply scale equalizers. In other words, we would say
>
> Nearest Neighbor < Bilnear Upsampling < Bilinear Upsampling with Scale Equalizers
>
> We hope that these points will convince you to target bilinear upsampling in our study.
>
> Finally, please check our revised manuscript, where changed or added parts are colored with blue. Thank you again for your valuable comments!

---

### Review · Reviewer_2ERv · 2024-06-10

**Summary Of Contributions:**

The paper addresses the problem of scale disequilibrium in multi-level feature fusion in semantic segmentation networks. The authors identify that bilinear upsampling, a common technique used in feature fusion, causes scale disequilibrium, leading to suboptimal training and performance. To address this, they propose using scale equalizers, which normalize feature scales across multiple levels after bilinear upsampling. The key contributions of this paper are:

1. **Identification of Scale Disequilibrium:** The paper highlights a universal issue in multi-level feature fusion due to bilinear upsampling, supported by theoretical and empirical evidence.

2. **Introduction of Scale Equalizers:** A novel solution is proposed to address the identified problem. The scale equalizers are hyperparameter-free, computationally inexpensive, and easy to implement.

3. **Experimental Validation:** The effectiveness of the proposed scale equalizers is validated across various datasets (ADE20K, PASCAL VOC 2012, and Cityscapes) and decoder architectures (UPerHead, PSPHead, ASPPHead, SepASPPHead, and FCNHead).

**Audience:**

Yes

**Claims And Evidence:**

Yes

**Requested Changes:**

1. **Comparison with Existing Normalization Techniques:** Include a detailed performance and computational overhead comparison between the proposed scale equalizers and other standard normalization techniques (e.g., batch normalization, layer normalization, instance normalization).

2. **Comprehensive Computational Time Analysis:** Conduct and present a thorough analysis of the computational time and resource usage associated with the proposed scale equalizers. Compare these findings with existing normalization techniques to substantiate claims about the method's efficiency.

3. **In-Depth Insights and Analysis:** Provide more visualization results and analysis into the reasons behind the performance improvements observed with the proposed method. Discuss potential limitations and scenarios where the method might be less effective. This will offer a more nuanced understanding of the method's strengths and weaknesses and guide future research directions.

**Strengths And Weaknesses:**

## Strengths

1. **Innovative Problem Identification:** The paper identifies and thoroughly investigates a subtle but impactful problem in multi-level feature fusion, which has not been addressed in previous research.

2. **Comprehensive Solution:** The proposed scale equalizer can be easily integrated into existing architectures without additional computational costs, making it practical for real-world applications.

3. **Robust Theoretical and Empirical Support:** The paper provides strong theoretical justification for the proposed method and supports its claims with empirical results across multiple settings.

## Weaknesses

1. **Limited Performance Gain:** The performance improvements reported in the paper are relatively modest, with increases in mIoU typically ranging from +0.1 to +0.4. While any improvement is beneficial, the incremental nature of these gains may not be compelling enough to justify the adoption of the proposed method in all scenarios.

2. **Lack of Comparison with Existing Normalization Techniques:** The paper does not provide a direct comparison between the proposed scale equalizers and other existing normalization techniques, such as batch normalization, layer normalization, or instance normalization. This omission makes it difficult to evaluate the relative effectiveness and advantages of the proposed method.

3. **No Analysis of Computational Time:** Although the paper claims that the proposed scale equalizers are computationally inexpensive at runtime and faster than traditional normalization methods, it does not include any detailed analysis of preprocessing to other normalization methods at runtime to support these claims. A thorough comparison of the computational time and resource usage would strengthen the argument for the method's efficiency compared to the performance gain.

4. **Limited Insights and Analysis:** The insights provided in the paper are relatively limited. Only the results of mIoU on different datasets and decoders are provided in the paper. The authors could enhance the depth of their analysis by providing more visualization and exploring the underlying reasons for the observed performance improvements, potential limitations, and scenarios where the proposed method might not be effective. This would provide a more nuanced understanding of the method's strengths and weaknesses.

---

> ### Author Response · Authors · 2024-08-07
> **Response to Reviewer 2ERv**
>
> Thank you for your insightful comments to improve the quality of this manuscript. We find that your comments are valid; in our revised manuscript, we reflected them as much as possible. Here, we provide a point-to-point response to your requested changes.
>
> > - Comparison with Existing Normalization Techniques: Include a detailed performance and computational overhead comparison between the proposed scale equalizers and other standard normalization techniques (e.g., batch normalization, layer normalization, instance normalization).
> > - Comprehensive Computational Time Analysis: Conduct and present a thorough analysis of the computational time and resource usage associated with the proposed scale equalizers. Compare these findings with existing normalization techniques to substantiate claims about the method's efficiency.
>
> Because these two points are connected, we answer them together. First, consider the pipeline used for generating each concatenation subject in multi-level feature fusion. The existing pipeline of [Conv--BatchNorm--ReLU--UP] yields concatenation subjects with different variances due to the last upsampling. Here, when modifying it into [Conv--ReLU--UP--BatchNorm], it outputs a normalized feature, which achieves an equalized scale across concatenation subjects. This behavior can also be achieved with other normalization layers, such as GroupNorm and LayerNorm.
>
> Although the use of a normalization layer after upsampling can be another solution to achieve scale equilibrium, it causes increased computational costs due to the enlarged size of the feature map. Because mean and std are computed for the incoming feature map, the larger size of the feature map requires much computational cost for computing the mean and std. Furthermore, the computed mean and std are used for normalization of each element, which leads to increased complexity overall.
>
> This behavior can be verified through simulation. Using $x \in R^{N \times C \times H \times W}$ for $N=16$ and $C=128$, we simulated $8 \times$ bilinear upsampling and compared the computation time required for the four pipelines: [Conv--BatchNorm--ReLU--UP], [Conv--ReLU--UP--BatchNorm], [Conv--ReLU--UP--GroupNorm], and [Conv--ReLU--UP--LayerNorm]. Figure 7 summarizes the results. We observed that the existing [Conv--BatchNorm--ReLU--UP] pipeline consistently exhibited faster computation, whereas modified pipelines showed significantly slower computation. Note that the computational cost in case of injecting scale equalizers is equal to the existing pipeline of [Conv--BatchNorm--ReLU--UP] because it can be implemented without extra cost (Section 3.2, Algorithm 1); therefore, in terms of computational complexity, we claim that our proposed method is superior compared with the use of a normalization layer.
>
> In addition to computational costs, we compared segmentation performance when using modified pipelines (Table 4). We empirically observed that modifying the existing ordering of [Conv--BatchNorm--ReLU--UP] has a side effect of degraded segmentation performance, which outweighs possible advantages. This phenomenon was consistently confirmed for BatchNorm, GroupNorm, and LayerNorm. In consideration of this, we opt for keeping the existing pipeline without modification in its ordering while injecting a scale equalizer at the end of the pipeline.
>
> After all, thank you for pointing out this issue. We believe that this analysis is worthy of being mentioned. In our revised manuscript, we added a Discussion section and mentioned this issue of comparison with other normalization layers.

---

> ### Author Response · Authors · 2024-08-07
> **Response to Reviewer 2ERv**
>
> > In-Depth Insights and Analysis: Provide more visualization results and analysis into the reasons behind the performance improvements observed with the proposed method. Discuss potential limitations and scenarios where the method might be less effective. This will offer a more nuanced understanding of the method's strengths and weaknesses and guide future research directions.
>
> Thank you for valuable comment. In our revised manuscript, Figure 8 provides segmentation examples for the ADE20K dataset. We find that injecting scale equalizers leads to a better understanding of the global context of images. Specifically, with scale equalizers, the global layout is better captured for large parts. Indeed, our analysis says that existing multi-level feature fusion suffers from scale disequilibrium, which lowers the contribution of the last feature map that contains rich global information. Here, injecting scale equalizers facilitates the last feature map to be involved in multi-level feature fusion, which leads to a better understanding of the global context of the image.
>
> Finally, please check our revised manuscript, where changed or added parts are colored with blue. Thank you again for providing careful checkpoints!

---

### Review · Reviewer_B3q3 · 2024-08-04

**Summary Of Contributions:**

The paper looks into the scale disequilibrium issue which is a problem with parallel deep neural networks that hurts their feature fusion performance. It impacts training performance with backpropagation (gradient descent).

The authors claim that scale disequilibrium is caused by bilinear upsampling. They show this is supported by both theoretical and empirical evidence.

Based on this observation, they propose injecting scale equalizers to achieve scale equilibrium across multi-level features after bilinear upsampling. Their proposed scale equalizers are essentially just normalization and therefore are easy to implement.
They empirically show that scale equalizers work across some datasets.

**Audience:**

Yes

**Broader Impact Concerns:**

No concerns.

**Claims And Evidence:**

Yes

**Requested Changes:**

Please address the following questions:

* How does this method differ from normalizations used in practice? like group norm and layer norm seem to solve all issues that come up in training.

* The formulation looks just like batch norm. Could you please clarify, in a network with batch norm, why your method is needed and exactly how to use it?

* What are other scenarios other than empirical examples in this paper where scale equalization matters?How can you justify the importance of your work in the community?

* Would scale equalization be an issue in big networks such as ViT?

* What's the trade off of computation versus performance gain when using your method?

**Strengths And Weaknesses:**

Strength:
- The proposed equalizer is a normalization and therefore is easy and architecture independent.
- The paper provides theoretical justification for the proposed approach.

Weaknesses:
- The paper fails to compare to other normalizations.
- It doesn't clearly discuss why this work is needed on top of batch norm,  specifically, in pager 10 where it tries to mention it, use of former and latter, completely confuses the reader.
- Empirical improvement is limited and is for special case of image segmentation.

---

> ### Author Response · Authors · 2024-08-07
> **Response to Reviewer B3q3**
>
> Thank you for your valuable comments to improve the quality of this manuscript. We find that your comments are valid; in our revised manuscript, we reflected them as much as possible. Here, we provide a point-to-point response to your requested changes.
>
> > - How does this method differ from normalizations used in practice? like group norm and layer norm seem to solve all issues that come up in training.
> > - The formulation looks just like batch norm. Could you please clarify, in a network with batch norm, why your method is needed and exactly how to use it?
> > - What's the trade off of computation versus performance gain when using your method?
>
> Because these three points are connected, we answer them together. First, consider the pipeline used for generating each concatenation subject in multi-level feature fusion. The existing pipeline of [Conv--BatchNorm--ReLU--UP] yields concatenation subjects with different variances due to the last upsampling. Here, when modifying it into [Conv--ReLU--UP--BatchNorm], it outputs a normalized feature, which achieves an equalized scale across concatenation subjects. This behavior can also be achieved with other normalization layers, such as GroupNorm and LayerNorm.
>
> Although the use of a normalization layer after upsampling can be another solution to achieve scale equilibrium, it causes increased computational costs due to the enlarged size of the feature map. Because mean and std are computed for the incoming feature map, the larger size of the feature map requires much computational cost for computing the mean and std. Furthermore, the computed mean and std are used for normalization of each element, which leads to increased complexity overall.
>
> This behavior can be verified through simulation. Using $x \in R^{N \times C \times H \times W}$ for $N=16$ and $C=128$, we simulated $8 \times$ bilinear upsampling and compared the computation time required for the four pipelines: [Conv--BatchNorm--ReLU--UP], [Conv--ReLU--UP--BatchNorm], [Conv--ReLU--UP--GroupNorm], and [Conv--ReLU--UP--LayerNorm]. Figure 7 summarizes the results. We observed that the existing [Conv--BatchNorm--ReLU--UP] pipeline consistently exhibited faster computation, whereas modified pipelines showed significantly slower computation. Note that the computational cost in case of injecting scale equalizers is equal to the existing pipeline of [Conv--BatchNorm--ReLU--UP] because it can be implemented without extra cost (Section 3.2, Algorithm 1); therefore, in terms of computational complexity, we claim that our proposed method is superior compared with the use of a normalization layer.
>
> In addition to computational costs, we compared segmentation performance when using modified pipelines (Table 4). We empirically observed that modifying the existing ordering of [Conv--BatchNorm--ReLU--UP] has a side effect of degraded segmentation performance, which outweighs possible advantages. This phenomenon was consistently confirmed for BatchNorm, GroupNorm, and LayerNorm. In consideration of this, we opt for keeping the existing pipeline without modification in its ordering while injecting a scale equalizer at the end of the pipeline.
>
> After all, thank you for pointing out this issue. We believe that this analysis is worthy of being mentioned. In our revised manuscript, we added a Discussion section and mentioned this issue of comparison with other normalization layers.

---

> ### Author Response · Authors · 2024-08-07
> **Response to Reviewer B3q3**
>
> > What are other scenarios other than empirical examples in this paper where scale equalization matters?How can you justify the importance of your work in the community?
>
> Thank you for your valuable comment. First, we find that the use of upsampling and multi-level feature fusion is quite prevalent in the machine learning community. Although we focused on the multi-level feature fusion in semantic segmentation networks as a prime example, our scale equalization would matter for other multi-level feature fusion tasks. Specifically, our scale equalizers generally matter for modern encoder-decoder networks. For example, monocular depth estimation networks have used the encoder-decoder architecture, whose multi-level feature fusion exhibits scale disequilibrium similar to the semantic segmentation networks.
>
> In consideration of this, we additionally verified scale equlization for a monocular depth estimation network. The target model was GEDepth, where its feature fusion module concatenates upsampled and non-upsampled feature maps. We compared performance before and after injecting scale equalizers into the feature fusion module (Table 5). We observed that injecting scale equalizers improved the performance of the monocular depth estimation task.
>
> In summary, we believe that scale equalization matters for other tasks where encoder-decoder architecture is deployed. In the revised manuscript, to support the general applicability of the scale equalizer, we mentioned this point by providing the additional experiment on monocular depth estimation.
>
> > Would scale equalization be an issue in big networks such as ViT?
>
> The scale disequlibrium arises within the decoder at the concatenation layer after upsampling. Therefore, this problem is related to the architectural design of the decoder; we believe that using a larger encoder network, albeit having much expressivity, cannot solve this problem. For example, our analysis in Equations 5-6 does not require a specific condition on feature x because the feature is subsequently normalized by batch normalization. In other words, regardless of property in feature x, the scale disequillibrium arises.
>
> Indeed, we tried our best to verify the effect of scale equalizer across a variety of models. In Table 2, we verified modern backbones from tiny to large: Swin-{T, S, B}, Twins-SVT-{S, B, L}, and ConvNeXt-{T, S, B}, where we observed that scale equalizer matters for all these backbones.
>
> Nevertheless, we think that clarifying this point would improve the quality of this manuscript. In our revised manuscript, we mentioned this point.
>
> Finally, please check our revised manuscript, where changed or added parts are colored with blue. Thank you again for thoughtful comments!

---

### Decision · Action_Editor_nhjd · 2024-10-31

**Recommendation:** Accept with minor revision

**Comment:**

please see above in claims section or the required edits.

**Audience:**

This paper is of interest in ML community especially the people who focus on normalization methods in the vision domain.

**Claims And Evidence:**

The paper looks into scale equalization claiming that _they_ have found that multi-level features from parallel branches are on different scales. They discuss that this is an unwanted state and propose a method to resolve it. Then they show in experiments that the proposed method works. In the paper the focus is specifically on semantic segmentation task.

There are a number of things to consider in the above claim.
First the fact that "that multi-level features from parallel branches are on different scales" has been known before and it is not specific to this paper. This needs to be corrected.

The technical method proposed by the author is a _universal solution to the imbalance problem commonly found in visual tasks_. However, the experiments conducted by the authors were only carried out in segmentation tasks, which contradicts the original intention of addressing this challenge. Either the claim needs to be updated or the experiments needs to be expanded.

The experimental results demonstrate that the proposed scale equalizer, both the original form in the network and the efficient version via initialization, can increase the mIoU index by about +0.1 to +0.4 across several datasets and architectures. However, the performance gain is relatively limited, especially compared to other normalization methods in Table 4 of the new manuscript. The suggested normalization method does not differ in nature in many other normalization methods that already exist and show great performance in giant networks. From the last three rows, we can see that replacing BatchNorm with GroupNorm and LayerNorm can lead to a gain of around 4 mIoU, respectively. In contrast, adding the proposed ScaleEQ only has a gain of 0.2 mIoU. Though the original pipeline with BatchNorm is somewhat faster than others, I think the authors should focus more on the reason why "BatchNorm --> OtherNorm" contributes more than "BatchNorm + ScaleEQ" and give a more detailed explanation.

---

> ### Author Response · Authors · 2024-11-11
> **Response to Action Editor**
>
> Thank you for providing careful checkpoints and the affirmative decision! Following your comments, we prepared a revised manuscript for the camera-ready version. Here, we provide a point-to-point response to your suggestion.
>
> > First the fact that "that multi-level features from parallel branches are on different scales" has been known before and it is not specific to this paper. This needs to be corrected.
>
> Your understanding is correct. Our finding is the root cause of the scale disequilibrium, not the phenomenon of scale disequilibrium itself. We corrected the claim in the abstract.
>
> > The technical method proposed by the author is a universal solution to the imbalance problem commonly found in visual tasks. However, the experiments conducted by the authors were only carried out in segmentation tasks, which contradicts the original intention of addressing this challenge. Either the claim needs to be updated or the experiments needs to be expanded.
>
> Thank you for your careful reading. As you commented, our main target is on multi-level feature fusion for semantic segmentation tasks. We updated this point in the introduction and conclusion.
>
> > From the last three rows, we can see that replacing BatchNorm with GroupNorm and LayerNorm can lead to a gain of around 4 mIoU, respectively. In contrast, adding the proposed ScaleEQ only has a gain of 0.2 mIoU. Though the original pipeline with BatchNorm is somewhat faster than others, I think the authors should focus more on the reason why "BatchNorm --> OtherNorm" contributes more than "BatchNorm + ScaleEQ" and give a more detailed explanation.
>
> Thank you for your valuable comment. In the last three rows of Table 4, we intended to emphasize that the three normalizations failed to exceed the baseline. Although GroupNorm or LayerNorm might yield improved performance compared with BatchNorm, their performances were even below the baseline performance of the existing pipeline of [Conv--BatchNorm--ReLU--UP]. Furthermore, applying a normalization layer after an upsampled feature requires much computational cost. Considering both computational cost and segmentation performance, the best pipeline is our proposed pipeline of [Conv--BatchNorm--ReLU--UP--ScaleEQ]. We added a more detailed explanation of Table 4 in Section 5.1.
>
> Again, thank you for your valuable comments! We feel that these clarifications further improved our manuscript. Please check the revised manuscript for the camera-ready version. If there are further requests to improve this manuscript, feel free to share your ideas with us.